# Loss of CTLH component MAEA impairs DNA repair and replication and leads to developmental delay

Søren H Hough [1,2,23], Satpal S Jhujh [3,23], Samah W Awwad[1,2,24], Oliver E Lewis[1,24], Simon Lam [1,2], John C Thomas [2], Thorsten Mosler [4], Aldo Bader[1,2], Lauren Bartik[5,6], Shane McKee[7], Shivarajan Amudhavalli[6,8], Estelle Colin[9,10], Nadirah Damseh[11], Emma Clement [12], Pilar Cacheiro [13], Anirban Majumdar[14], Damian Smedley[13], Joël Fluss[15], Rosalinda Giannini[16], Isabelle Thiffault[8,17,18], Guido Zagnoli Vieira [2], Rimma Belotserkovskaya[1,2], Stephen J Smerdon [3], Petra Beli[4,19], Yaron Galanty [1,2✉], Christopher J Carnie [1,2,20,21,22✉], Grant S Stewart [3✉] & Stephen P Jackson [1,2✉]

## Abstract

Ubiquitin E3 ligases play crucial roles in the DNA damage response (DDR) by modulating the turnover, localization, activation, and interactions of DDR and DNA replication proteins. We performed a CRISPR-Cas9 knockout screen focused on ubiquitin E3 ligases and related proteins with the DNA topoisomerase I inhibitor camptothecin. This led us to establish that MAEA, a core subunit of the CTLH E3 ligase complex, is a critical regulator of homologous recombination and the replication stress response. In tandem, we identified eight patients with variants in *MAEA* who present with a neurodevelopmental disorder that we term DIADEM (Developmental delay and Intellectual disability Associated with DEfects in MAEA). Analysis of patient-derived cell lines and mutation modeling reveal an underlying defect in HR-dependent DNA repair and replication fork restart and protection as a likely cause of disease. Mechanistically, we find that MAEA dysfunction hinders DNA repair by reducing the efficiency of RAD51 loading at sites of DNA damage, which we propose may contribute to the presentation of DIADEM by compromising genome integrity and cell division during development.

Keywords DNA Repair; DNA Replication; Ubiquitin; Neurodevelopmental Disorder
Subject Category Genetics, Gene Therapy & Genetic Disease

## Introduction

As DNA is constantly attacked by exogenous and endogenous agents, DNA repair pathways must function efficiently to avoid the accumulation of deleterious genomic alterations (Jackson and Bartek, 2009). There is ample crosstalk between the DNA replication and DNA repair processes, exemplified by RAD51, which is essential for homologous recombination (HR)-dependent repair of DNA double-strand breaks (DSBs) and protection/restart of damaged replication forks (Zellweger et al, 2015; Bhat and Cortez, 2018; Liu et al, 2023). Consequently, germline loss-of-function or hypomorphic mutations in DNA damage response (DDR) genes often compromise both DNA repair and replication, leading to clinical phenotypes including neurodegeneration, immunodeficiency, skeletal abnormalities, intellectual dysfunction, bone marrow failure, and growth delay (Jackson and Bartek, 2009; Qing et al, 2023).

Protein ubiquitylation is critical in promoting the recruitment and retention of proteins to sites of DNA damage, regulating DNA repair and replication protein turnover, and/or altering their enzymatic activities (Jackson and Durocher, 2013; Brown and

[1]Cancer Research UK Cambridge Institute, Li Ka Shing Building, Robinson Way, Cambridge CB2 0RE, UK. [2]The Gurdon Institute and Department of Biochemistry, University of Cambridge, Tennis Court Road, Cambridge CB2 1QN, UK. [3]Institute of Cancer and Genomic Sciences, College of Medical and Dental Sciences, University of Birmingham, Birmingham, UK. [4]Institute of Molecular Biology (IMB), Chromatin Biology & Proteomics, Mainz, Germany. [5]Department of Pediatrics, Division of Clinical Genetics, Children's Mercy Hospital, Kansas City, MO, USA. [6]Kansas City School of Medicine, University of Missouri, Kansas City, MO, USA. [7]Northern Ireland Regional Genetics Service, Belfast City Hospital, Belfast BT9 7AB, UK. [8]Department of Pathology & Genetics, Children's Mercy Hospital, Kansas City, MO, USA. [9]Service de Génétique Médicale, CHU d'Angers, Angers, France. [10]Université Angers, [CHU Angers], INSERM, CNRS, MITOVASC, SFR ICAT, F-49000 Angers, France. [11]Department of Pediatrics, Makassed Hospital and Al-Quds University, East Jerusalem, Palestine. [12]Great Ormond Street Hospital for Children NHS Foundation Trust, London, UK. [13]William Harvey Research Institute, School of Medicine and Dentistry, Queen Mary University of London, London, UK. [14]Department of Paediatric Neurology, Bristol Children's Hospital, Bristol, UK. [15]Pediatric Neurology Unit, University Children's Hospital Geneva, Geneva, Switzerland. [16]Division of Medical Genetics, Diagnostics Department, Geneva University Hospitals, Geneva, Switzerland. [17]Genomic Medicine Center, Children's Mercy Hospital and Research Institute, Kansas City, MO, USA. [18]Faculty of Medicine, University of Missouri-Kansas City, Kansas City, MO, USA. [19]Institute of Developmental Biology and Neurobiology (IDN), Johannes Gutenberg-Universität, Mainz, Germany. [20]Molecular Medicine Partnership Unit (MMPU), Heidelberg University and European Molecular Biology Laboratory (EMBL), Heidelberg, Germany. [21]Department of Pediatric Oncology, Hematology and Immunology, Heidelberg University Hospital, Heidelberg, Germany. [22]Hopp Children's Cancer Center (KiTZ), Heidelberg, Germany. [23]These authors contributed equally: Søren H Hough, Satpal S Jhujh. [24]These authors contributed equally: Samah W Awwad, Oliver E Lewis. ✉E-mail: yaron.galanty@cruk.cam.ac.uk; christopherjames.carnie@med.uni-heidelberg.de; g.s.stewart@bham.ac.uk; steve.jackson@cruk.cam.ac.uk

Jackson, 2015; Polo and Jackson, 2011). Ubiquitylation in human cells is tightly regulated and mediated via enzymatic cascades involving either one of two E1 activating enzymes, one of ~40 E2 conjugating enzymes, and over 600 known E3 ligases, and various associated factors such as substrate adapters that enhance target specificity (Brown and Jackson, 2015; Schmidt et al, 2015; Chauhan et al, 2024). Multiple E3 ubiquitin ligases, including those involved in the DDR, have previously been linked to inherited and de novo pathologies in humans associated with diverse clinical phenotypes. These include immunodeficiency and neurodegeneration (e.g., RNF168); growth delay and cancer predisposition (e.g., BRCA1); neurodevelopmental abnormalities (e.g., RFWD3); and primordial dwarfism (e.g., TRAIP) (Stewart et al, 2007, 2009; Domchek et al, 2013; Sawyer et al, 2015; Knies et al, 2017; Chauhan et al, 2024).

Here, using CRISPR-Cas9 screening (Awwad et al, 2023), we identify the CTLH (C-terminal to LisH) ubiquitin E3 ligase complex as a regulator of DNA repair and replication. We demonstrate that CTLH—and specifically its RING domain-containing MAEA subunit required for catalytic activity—promotes RAD51 loading at DNA damage sites. We also show that loss of MAEA severely impairs HR, as well as replication fork progression, protection, and restart. Furthermore, we describe a cohort of patients with pathogenic MAEA variants who exhibit neurodevelopmental defects and abnormalities in the cellular replication stress response. This highlights the clinical importance of the CTLH complex in maintaining genome instability.

# Results

## Loss of the CTLH complex sensitizes cells to seDSB-inducing agents

To identify ubiquitylation components mediating responses to single-ended DNA DSBs (seDSBs), we generated a focused single guide RNA (sgRNA) CRISPR-Cas9 knockout library targeting 886 E3 ligases and associated substrate adapters and used this to screen for genes that impact the sensitivity of U2OS cells to the topoisomerase I (TOP1) inhibitor, camptothecin (Fig. 1A; Table EV1). Camptothecin-induced seDSBs are S-phase specific and repaired via HR (Caldecott, 2024). The screen successfully identified E3 ubiquitin ligases with known roles in HR, including RNF168, BRCA1, and BARD1 (Fig. 1B; Dataset EV1). The CRISPR screen also identified subunits of the CTLH complex as conferring resistance to camptothecin. These included MAEA, WDR26, RMND5A, GID8, and RANBP9 (Fig. 1B; Dataset EV1), which represented every CTLH subunit in our focused sgRNA library.

The CTLH complex is known to regulate gluconeogenic enzymes in yeast (Menssen et al, 2012) and glycolytic enzymes in mammals (Maitland et al, 2021). However, our results suggested the complex has a previously unrecognized role in the DDR. Given that the ubiquitylation activity of the CTLH complex depends on the functional RING domain of the catalytic subunit MAEA (Santt et al, 2008; Braun et al, 2011), we generated MAEA knockout (KO) and hypomorphic (HM) U2OS cells using CRISPR-Cas9 (Figs. 1C and EV1A–C). The HM allele is a 33-base pair in-frame deletion leading to loss of amino acid residues 108–117, resulting in a truncated protein expressed at lower levels than wild-type (WT) MAEA (Fig. EV1A). Compared to WT cells, MAEA KO and HM

cells were hypersensitive to camptothecin and the PARP inhibitor, olaparib—both of which induce seDSBs during S phase (Fig. 1D). We found that inactivating the genes encoding CTLH components MAEA, WDR26, RMND5A, and RANBP9 led to camptothecin and olaparib hypersensitivity in HAP1 cells, demonstrating that this role is shared with other CTLH subunits and is not a U2OS-specific effect (Fig. 1E). Goh et al recently showed that HAP1 cells are unusually sensitive to camptothecin due a debilitating mutation in TDP1 (Goh et al, 2025). Further sensitization of HAP1 cells to camptothecin upon CTLH loss suggested that TDP1 and CTLH function in parallel pathways to protect cells from abortive TOP1 lesions.

MAEA loss did not confer detectable hypersensitivity to etoposide (Fig. 1F), which induces double-ended DSBs that do not primarily rely on HR for repair. Thus, our findings suggested that CTLH may function specifically in HR-related events during S phase. We observed that MAEA loss increased the percentage of cells in S phase, supporting the idea that these cells experience replication stress even without exposure to exogenous genotoxic agents (Fig. EV1D). In support of this notion, we observed that MAEA-deficient U2OS cells form smaller colonies than WT cells, although this phenotype does not appear to be a feature of MAEA KO HAP1 cells (Fig. EV1E,F). We also targeted MAEA using siRNA (Fig. EV1G) in U2OS cells to ensure the observed phenotypes were not the result of adaptation in knockout cells. We assessed plating efficiency in untreated conditions and observed that siRNA-mediated MAEA depletion compromised both colony formation and growth (Fig. EV1H–K).

## Clinical MAEA variants associated with neurodevelopmental defects in humans hypersensitize cells to seDSB-inducing agents

The CTLH complex functions in development (Goto and Shibuya, 2022; Briney et al, 2025), and while mutations in the core CTLH complex have not been previously associated with pathogenesis, variants in the CTLH substrate adapter, WDR26, can cause Skraban-Deardorff (SD) syndrome. SD syndrome is characterized by a developmental delay, abnormal gait, and seizures (Skraban et al, 2017; Pavinato et al, 2021; Cospain et al, 2021; Cheng et al, 2022; Hu et al, 2022; Innella et al, 2023). SD has been associated with compromised CTLH complex assembly and chromatin accessibility (Gross et al, 2024; Onea et al, 2025).

From the Deciphering Developmental Disorders (DDD) database and the 100,000 Genomes Project, and by establishing clinical connections via GeneMatcher, we identified eight individuals with likely pathogenic variants in MAEA (Tables EV2 and 3) (Firth et al, 2009; Turnbull et al, 2018; Sobreira et al, 2015; Cacheiro et al, 2020). All individuals presented with developmental delay, intellectual disability, and delayed acquisition of speech. All patients (three male and five female) were between the ages of 19 months and 16 years at the time of evaluation, and consistent traits included delayed speech and language acquisition, developmental delay, and intellectual disability (DD/ID). Most (7/8) had abnormal muscle tone, dysmorphic features, and motor delay. Seizures and autism spectrum disorder were found in a subset (2/8 and 1/8, respectively). In summary, the characteristics of these patients may constitute a novel nonsyndromic DD/ID. While most patients (7/8) exhibited some clinical features typically associated

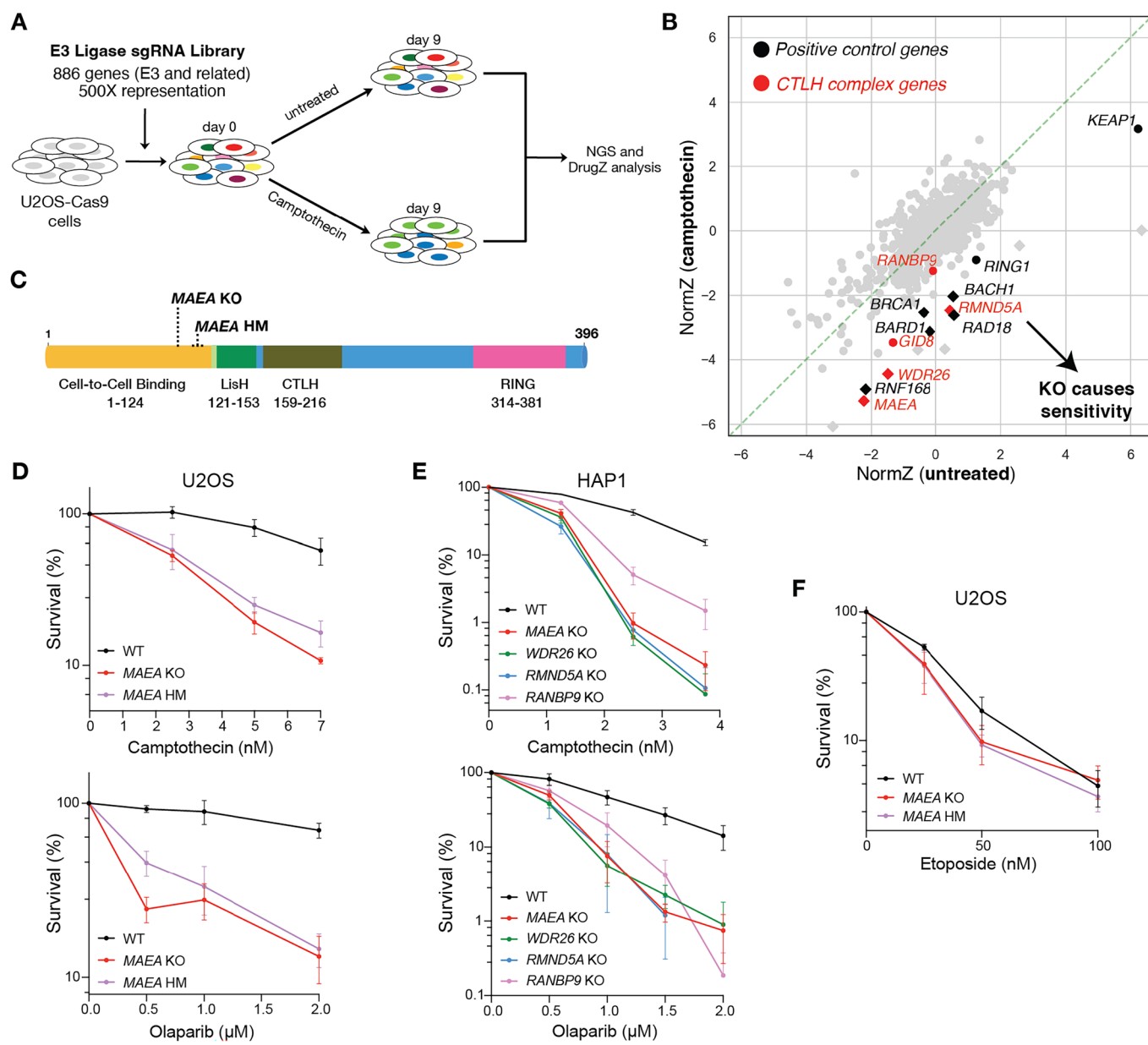

**Figure 1. Loss of CTLH components confers hypersensitivity to seDSB-inducing agents.**

(A) Overview of the E3 ubiquitin ligase CRISPR screen. (B) DrugZ analysis of (A). Black symbols denote positive control proteins, while red symbols denote CTLH complex members. Diamonds and circles denote significant and non-significant hits, respectively. (C) MAEA domain map displaying CRISPR-induced edits in *MAEA* KO and *MAEA* hypomorph U2OS cells. (D) Clonogenic survival assays in *MAEA* KO U2OS cells using camptothecin and olaparib. (E) Clonogenic survival assays in HAP1 cell lines using camptothecin and olaparib. (F) Clonogenic survival assays using etoposide in the indicated U2OS cell lines. (D–F) *n* = 3 independent experiments; error bars denote mean ± SEM. KO knockout, HM hypomorph. Source data are available online for this figure.

with known inherited DNA repair or replication deficiency disorders, others, such as seizures, short stature, moderate-to-severe microcephaly, skeletal abnormalities, behavioral issues, immunodeficiency, and bone marrow failure, were either absent or displayed only by one or two affected individuals (Tables EV2 and 3).

All but one of the identified patient-associated *MAEA* variants were de novo heterozygous variants, suggesting that the affected allele has dominant-negative effects. The other variant (c.1009 C >

T;p.Arg337Cys), present in two affected siblings (MAEA-P5a and MAEA-P5b), was present in a homozygous state suggestive of a recessive mode of inheritance. These siblings also inherited a homozygous variant of uncertain significance (c.317 G > A;pArg106His) in *LETM1*, a gene previously linked with a neurodegenerative disorder caused by mitochondrial dysfunction (Kaiyrzhanov et al, 2022). However, a comparison of the patients across the cohort coupled with assessments of lactate levels, urine organic acids, and plasma amino acids in patients P5a and P5b indicated

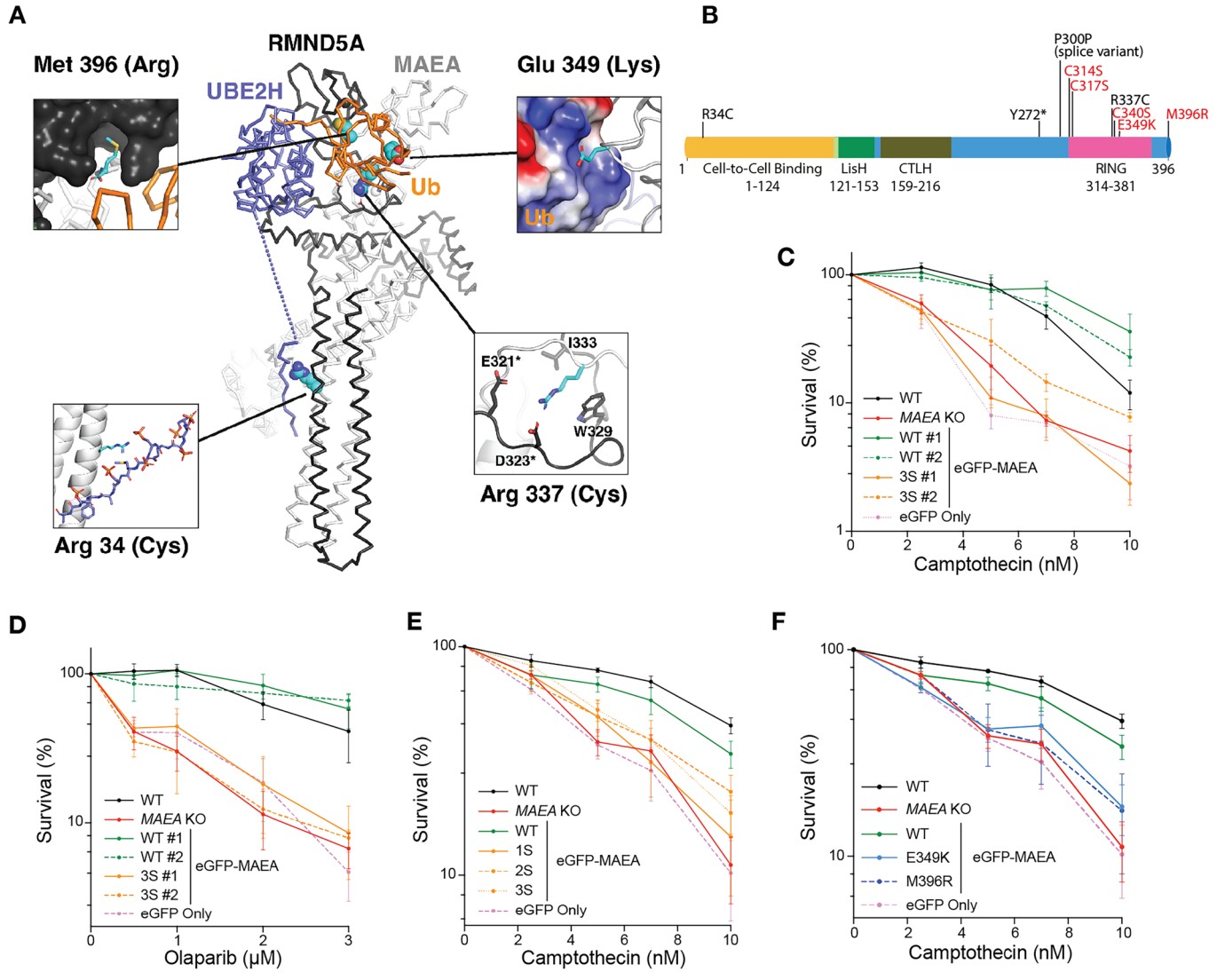

**Figure 2. Clinical *MAEA* variants associated with neurodevelopmental defects in humans hypersensitize cells to seDSB-inducing agents.**

(A) Cα trace representation of the MAEA/RMND5A/UBE2H complex crystal structure (PDBID: 8PJN). Altered residues are shown as CPK spheres colored by atom (C – cyan, O – red, N – blue, S – yellow). The environments of each mutant site are shown in the expansion insets. (B) Domain map of MAEA with C > S mutations and clinical variants labeled. Red text indicates cells were complemented with these MAEA variants and tested in subsequent assays. (C–F) Clonogenic survival assays in the indicated U2OS cell lines using camptothecin or olaparib, as indicated. Data in (C–F) are the combined results of three independent experiments. Bars denote mean ± SEM. *$P \le 0.05$, **$P \le 0.01$, ***$P \le 0.001$, ****$P \le 0.0001$. 1S = eGFP-MAEA$^{1S}$ (C340S), 2S = eGFP-MAEA$^{2S}$ (C314S, C317S), 3S = eGFP-MAEA$^{3S}$ (C314S, C317S, and C340S). Source data are available online for this figure.

that the *LETM1* variant was unlikely to be pathogenic. Moreover, we have identified the same *LETM1* variant in another patient (with no *MAEA* variant) who has no cognitive impairment, further suggesting that *LETM1* dysfunction is unrelated to the DD/ID exhibited by patients P5a/P5b. In summary, the characteristics observed in these eight patients likely constitute a novel nonsyndromic DD/ID associated with pathogenic variants in *MAEA*, which we have termed DIADEM (**D**evelopmental delay and **I**ntellectual disability **A**ssociated with **DE**fects in **M**AEA).

Further analyses suggested that all patient-associated MAEA variants we describe are pathogenic. All missense/truncating MAEA variants except two (those in P2 and P6) are in or near its C-terminal RING domain. The intronic variant present in P2 is

predicted to disrupt splicing and create a frameshift mutation that truncates MAEA just before the RING domain. To better understand and predict the impacts of these patient variants, we used existing protein structures (Chrustowicz et al, 2023) to interrogate the local protein-structural environments of each of the four MAEA patient missense variants within the context of the CTLH catalytic module containing MAEA, RMND5A, UBE2H, and ubiquitin (Fig. 2A) (Chrustowicz et al, 2023). This revealed that Met-396 packs into a hydrophobic pocket on the RMND5A surface that cannot accommodate a substitution with the more bulky and polar arginine side chain found in patient P4. By contrast, Glu-349 engages in electrostatic attraction with a region of positive potential on the ubiquitin surface, and substitution with lysine, as in patients

P1 and P7, introduces additional steric bulk and apposition of like charges.

Arg-337 engages in non-polar interactions with both tryptophan and valine residues from RMND5A, so replacement with cysteine, as in patients P5a and P5b, would reduce the extent of these favorable contacts and would also remove other favorable electrostatic interactions with glutamate and aspartate residues from the same RMND5A loop. Finally, the RING-distal substitution of Arg-34 with cysteine in patient P6 appears to directly affect a cluster of phosphorylation-dependent interactions between the MAEA/RMND5A stalk region and the UBE2H C-terminal region that are crucial for the formation of the core MAEA/RMND5A/UBE2H complex (Lampert et al, 2018) (Fig. 2A). Together, these analyses suggested that the patient-associated MAEA variants we identified would significantly compromise the integrity of the CTLH ubiquitin ligase complex.

To test whether MAEA ubiquitylation activity is required for its role in the DDR, we complemented *MAEA* KO U2OS cells with multiple MAEA constructs. These included WT N-terminal tagged eGFP-MAEA (eGFP-MAEA$^{WT}$) and a RING domain mutant (eGFP-MAEA$^{3S}$) version of eGFP-MAEA containing three cysteine to serine (C > S) substitutions: Cys314Ser, Cys317Ser, and Cys340-Ser (Figs. 2B and EV2A,B). We mutated these sites based on their evolutionary conservation and their predicted and observed importance for ubiquitylation: Cys314 and Cys317 match the RING domain consensus motif ($\mathbf{C}$-X2-$\mathbf{C}$-X[9-39]-$\mathbf{C}$-X[1-3]-$\mathbf{H}$-X[2–3]-$\mathbf{C}$- X2-$\mathbf{C}$-X[4-48]-$\mathbf{C}$-X2-$\mathbf{C}$) while Cys340 was reported to be essential for MAEA (Gid9) activity in yeast (Braun et al, 2011). The corresponding cysteine in the RING of RMND5A (Gid2) reportedly plays a similar role (Santt et al, 2008). We also generated eGFP-MAEA$^{1S}$ (Cys340Ser) and eGFP-MAEA$^{2S}$ (Cys314Ser, Cys317Ser) cell lines for comparison (Figs. 2B and EV2B).

In clonogenic survival assays, expression of eGFP-MAEA$^{WT}$ rescued the hypersensitivity of *MAEA* KO cells to camptothecin and olaparib, while expression of eGFP (hereafter eGFP Only) did not. Furthermore, eGFP-MAEA$^{1S}$, eGFP-MAEA$^{2S}$, and eGFP-MAEA$^{3S}$ variants were expressed at similar levels to eGFP-MAEA$^{WT}$, but failed to complement hypersensitivity to camptothecin and olaparib (Figs. 2C–E and EV2B). These investigations thus indicated that the RING domain—and therefore the ubiquitin ligase function—of MAEA is critical for cellular tolerance of seDSB induction.

We next utilized the *MAEA* KO U2OS cells to model the impact of MAEA variants from patients P1 and P7 (eGFP-MAEA$^{E349K}$) and P2 (eGFP-MAEA$^{M396R}$) in an isogenic cell background (Table EV2; Fig. 2B). In clonogenic survival assays, unlike eGFP-MAEA$^{WT}$, eGFP-MAEA$^{E349K}$ and eGFP-MAEA$^{M396R}$ failed to complement the camptothecin hypersensitivity of MAEA KO cells, despite expressing at similar levels to eGFP-MAEA$^{WT}$ (Figs. 2F and EV2B). This indicated that these clinical variants, like disruption of the MAEA RING domain, confer hypersensitivity to seDSB-inducing agents.

## MAEA loss impairs HR and RAD51 loading but not DNA end resection

To expand on the above findings, we used the "traffic light reporter" (TLR) assay to measure HR efficiency (Certo et al, 2011) in cells depleted of MAEA or RMND5A. In this system, accurate repair of the chromosomally integrated GFP target only occurs if the induced DSB is repaired by HR with a donor template. As expected, depletion of CtIP, which is crucial for DSB end resection (Huertas and Jackson, 2009), greatly impaired HR (Fig. 3A). Moreover, following siRNA-mediated depletion of MAEA or RMND5A, we observed a significant reduction (40–80%) of HR relative to cells transfected with a control siRNA against luciferase (Figs. 3A and EV3A; note that the analyses included adjustments for cell cycle profiles). Taken together, these findings indicated a key role for MAEA and the CTLH complex in promoting HR-dependent DNA repair.

We next investigated where CTLH functions in the HR pathway. DNA end resection is a decisive step during DNA DSB processing that dictates repair by HR. Therefore, we examined the levels of single-stranded DNA (ssDNA), as measured by both native BrdU staining and chromatin-bound RPA, in control and *MAEA* KO cells following camptothecin exposure (Sartori et al, 2007). Only cells that were γH2AX positive were included in our analysis, since camptothecin mainly induces DSBs (and therefore γH2AX signal) in S phase. We observed no resection defects in S phase *MAEA* KO cells after camptothecin treatment (Figs. 3B–E and EV3B–D). However, we observed a modest increase in native BrdU staining and chromatin-bound RPA levels in untreated *MAEA* KO S phase cells, which was rescued by CtIP depletion (Figs. 3B–E and EV3B–D). These data indicate that MAEA does not hinder camptothecin-induced DSB resection. They also suggested that MAEA loss leads to increased resection markers in untreated conditions, which most likely arise from defective HR-dependent resolution of endogenously occurring DNA damage at sites of DNA replication.

Following resection and RPA loading, a key next step in HR is the replacement of RPA with RAD51. We therefore quantified RAD51 foci formation, a known marker of functional HR (Naipal et al, 2014), in S phase WT and *MAEA* KO U2OS cells following camptothecin treatment, which both activate HR and induce RAD51-mediated replication fork reversal (Zellweger et al, 2015; Chaudhuri et al, 2012). Compared with WT cells, *MAEA* KO cells exhibited a pronounced failure to form RAD51 foci in S phase cells despite expressing RAD51 at WT levels (Figs. 3F–G and EV3E). We next assessed the expression of other factors relevant to DSB end resection and RAD51 loading by immunoblot. We observed no substantial impact of MAEA loss on levels of ATM, BRCA1, BRCA2, CtIP, DNA2, EXO1, or MRE11 (Fig. EV3F). Next, given the neurological presentation of DIADEM, we investigated resection and recombination markers in the neuronal-like neuroblastoma cell line SH-SY5Y. MAEA depletion in SH-SY5Y cells (Fig. EV4A) led to the same modest increase in BrdU and impaired RAD51 loading as observed in U2OS cells (Fig. EV4B–E). These data indicated that the HR deficiency conferred by MAEA loss is associated with an impaired RAD51 loading phenotype.

## MAEA loss leads to exacerbated replication stress

Camptothecin and olaparib cause replication stress as well as seDSBs (Zellweger et al, 2015). In unperturbed conditions, compared to WT cells, *MAEA* KO cells were enriched in the S/G2 phases of the cell cycle (Fig. EV1D) and exhibited elevated ssDNA (Figs. 3B–E and EV3B–D). Since these phenotypes were suggestive of endogenous replication stress, and since *MAEA* loss also sensitized cells to the ATR inhibitor (ATRi) AZD6738 in our previous study (Awwad et al, 2025), we hypothesized that MAEA

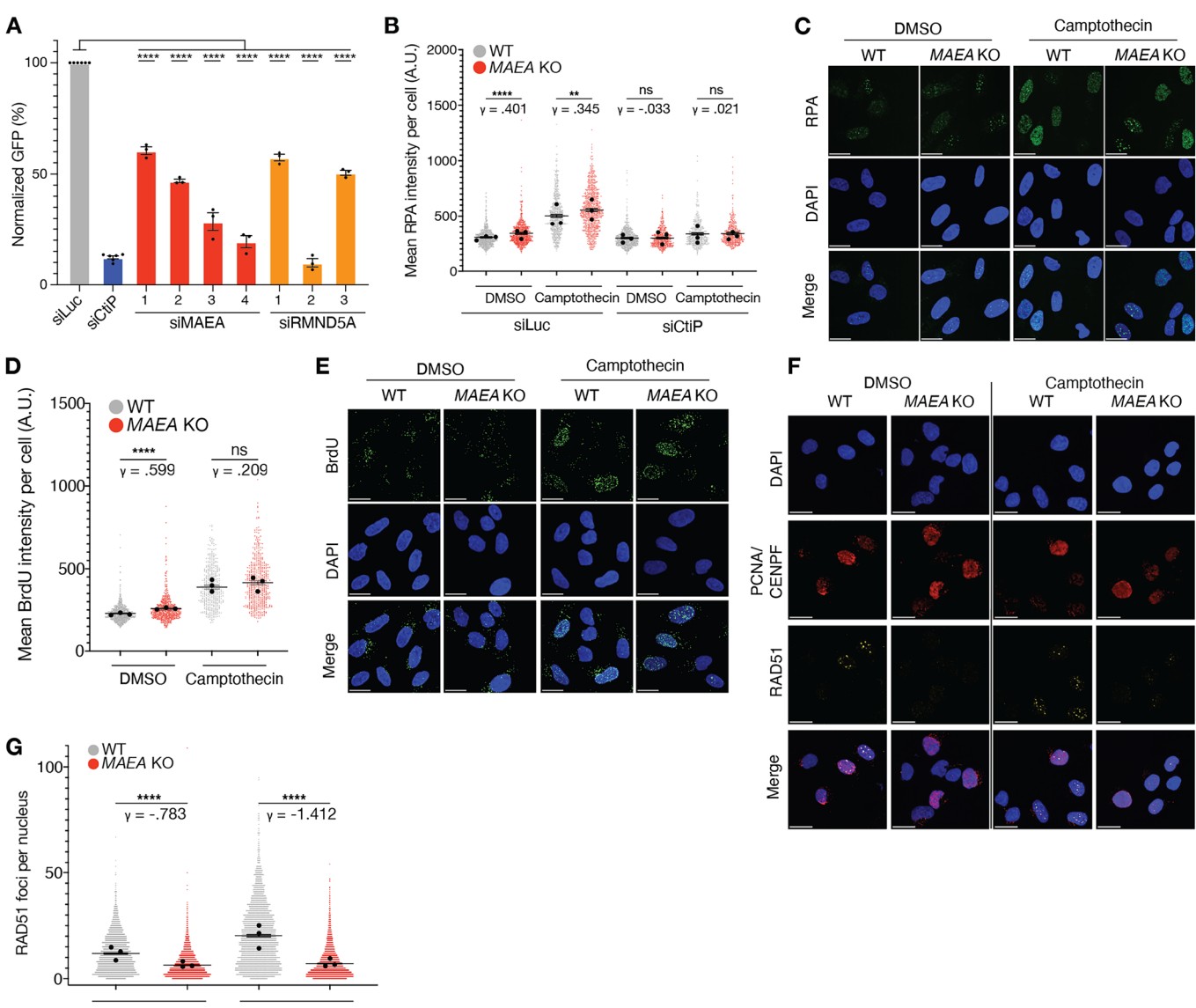

**Figure 3. MAEA loss impairs RAD51 loading, but not DNA end resection.**

(A) TLR assay in U2OS cells with the indicated siRNAs ($n = 3$). Statistics were generated by performing an ordinary one-way ANOVA comparing each siRNA to the siLuc control. siCtIP and siLuc data are from five independent experiments. siMAEA and siRMND5A data come from three independent experiments. (B, C) Immunofluorescence-based quantification (B) and representative images (C) of chromatinized RPA in S phase cells treated with DMSO (1 h) or camptothecin (1 μM, 1 h). (D, E) Quantification (D) and representative images (E) of BrdU in U2OS cells treated with DMSO (1 h) or camptothecin (1 μM 1 h). (F, G) Representative images (F) and quantification (G) of RAD51 foci in S phase U2OS cells treated with DMSO or camptothecin. All quantifications of foci are the combined results of three independent experiments. Extended fluorescent images from (C, E) can be found in Fig.EV3C,D. For (B–G), P values were generated using a two-tailed Kruskal–Wallis test. γ is a measure of effect size. Data represents three independent experiments. Bars denote mean ±95% CI. *$P \leq 0.05$, **$P \leq 0.01$, ***$P \leq 0.001$, ****$P \leq 0.0001$; exact P values can be found in Appendix Table S1. Scale bars = 20μm. Source data are available online for this figure.

loss might sensitize cells to other replication stress-inducing agents. Thus, we performed clonogenic survival assays with the replication stress-inducing agents hydroxyurea (HU), aphidicolin, and AZD6738. In all cases, we observed hypersensitivity of *MAEA* KO cells to these genotoxins compared with WT cells (Fig. 4A–C).

These data suggested that MAEA is required for effective HR at damaged replication forks. We investigated this by treating *MAEA* KO and WT U2OS cells with 0.5 mM HU, which primarily induces RAD51-dependent replication fork reversal rather than DSBs

(Zellweger et al, 2015). Thus, an increased γH2AX signal is likely the result of replication fork collapse. We measured the level of γH2AX in cells with replication fork stalling as marked by chromatin-bound RPA. This approach revealed that replicating cells lacking MAEA exhibit increased γH2AX accumulation in *MAEA* KO cells compared with WT cells (Fig. 4D). Moreover, *MAEA* KO cells exhibited increased γH2AX accumulation over time compared to WT cells upon ATR inhibition, especially at the later time point of 18 h (Fig. 4E). We used this time point to

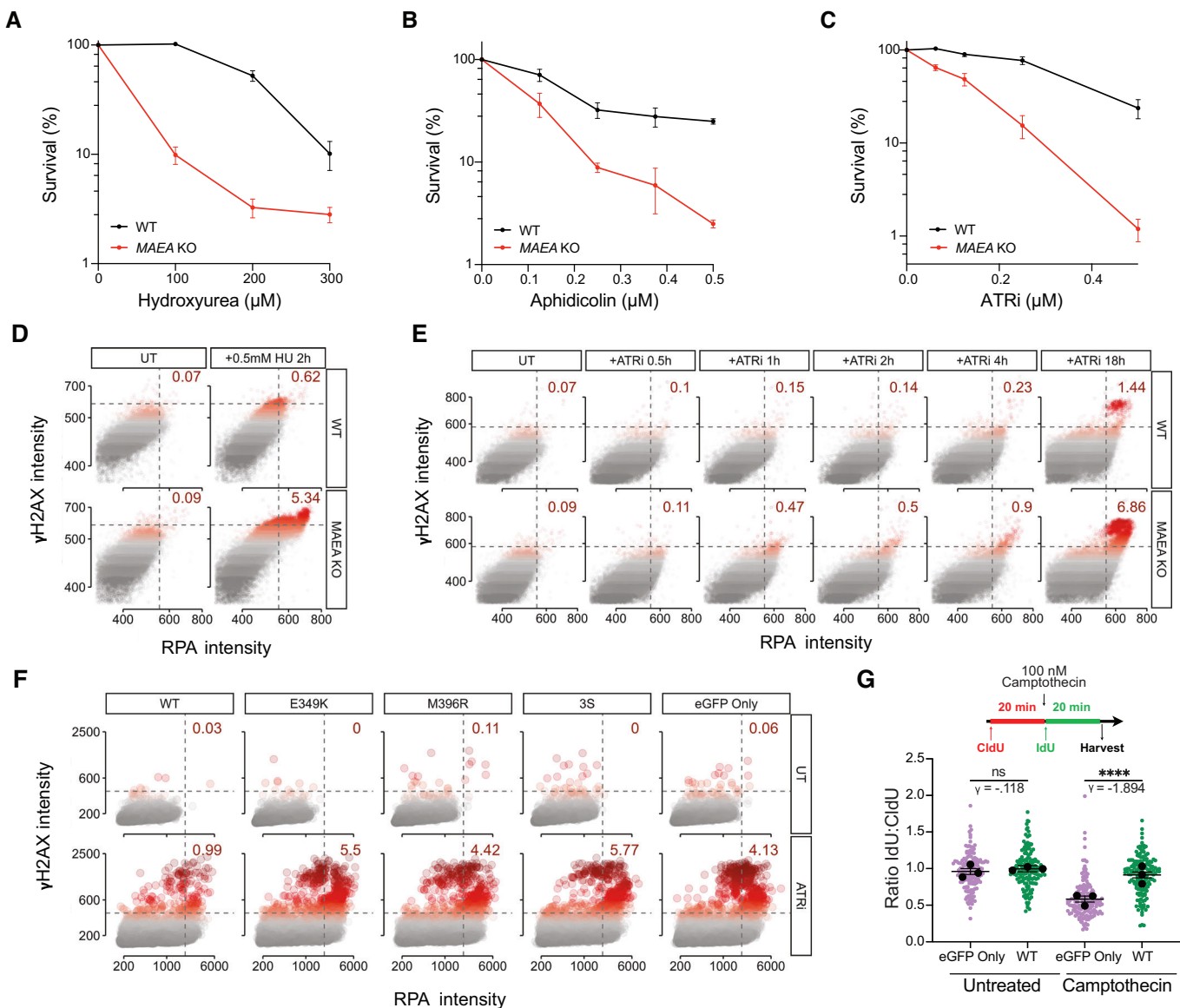

**Figure 4. MAEA KO cells are hypersensitive to replication stress.**

(A–C) Clonogenic survival assays in WT and *MAEA* KO cells with HU (A), aphidicolin (B), and ATRi (C). (D, E) Quantification of chromatin-associated γH2AX and RPA signal in WT and *MAEA* KO cells upon HU (D) or ATRi (E) treatment for the indicated times. (F) Quantification of chromatin-associated γH2AX and RPA signal upon ATRi treatment in *MAEA* KO U2OS cells complemented with the indicated eGFP expression constructs. Representative images are shown in Fig. EV4F. (G) DNA fiber assay measuring replication fork progression in eGFP Only (*MAEA*⁻/⁻) and eGFP-MAEA^WT cells following treatment with camptothecin. Clonogenic data (A–C) are from three independent experiments. Statistics were generated using an ordinary two-way ANOVA. Bars represent the mean ± SEM. DNA fiber data are the combined result of three independent experiments. *P* values were generated by performing a two-tailed Kruskal–Wallis test. γ is a measure of effect size. Bars represent the mean ± 95% CI. *$P \leq 0.05$, **$P \leq 0.01$, ***$P \leq 0.001$, ****$P \leq 0.0001$; exact *P* values can be found in Appendix Table S1. Scatter plots represent two independent experiments. HU hydroxyurea, ATRi ATR inhibitor (AZD3768). Source data are available online for this figure.

measure the accumulation of γH2AX in *MAEA* KO cells complemented with eGFP-MAEA^WT, eGFP-MAEA^E349K, eGFP-MAEA^M396R, eGFP-MAEA^3S, or eGFP-only. In accordance with results from the clonogenic survival assays using camptothecin and olaparib in these cell lines (Fig. 2C–F), we observed an increase in γH2AX levels in the variants and null cell lines compared with eGFP-MAEA^WT cells following ATRi treatment (Figs. 4F and EV4). These data implied that the RING domain-dependent E3 ubiquitin ligase function of MAEA is important in managing replication

stress and that clinical MAEA variants confer hypersensitivity to replication stress-inducing agents. Collectively, our findings indicated that MAEA/CTLH deficiency compromises cellular tolerance of seDSBs and replication stress.

## MAEA protects and promotes DNA replication

To elucidate potential effects of MAEA loss on DNA replication, we used DNA fiber spreading assays (Nieminuszczy et al, 2016). We

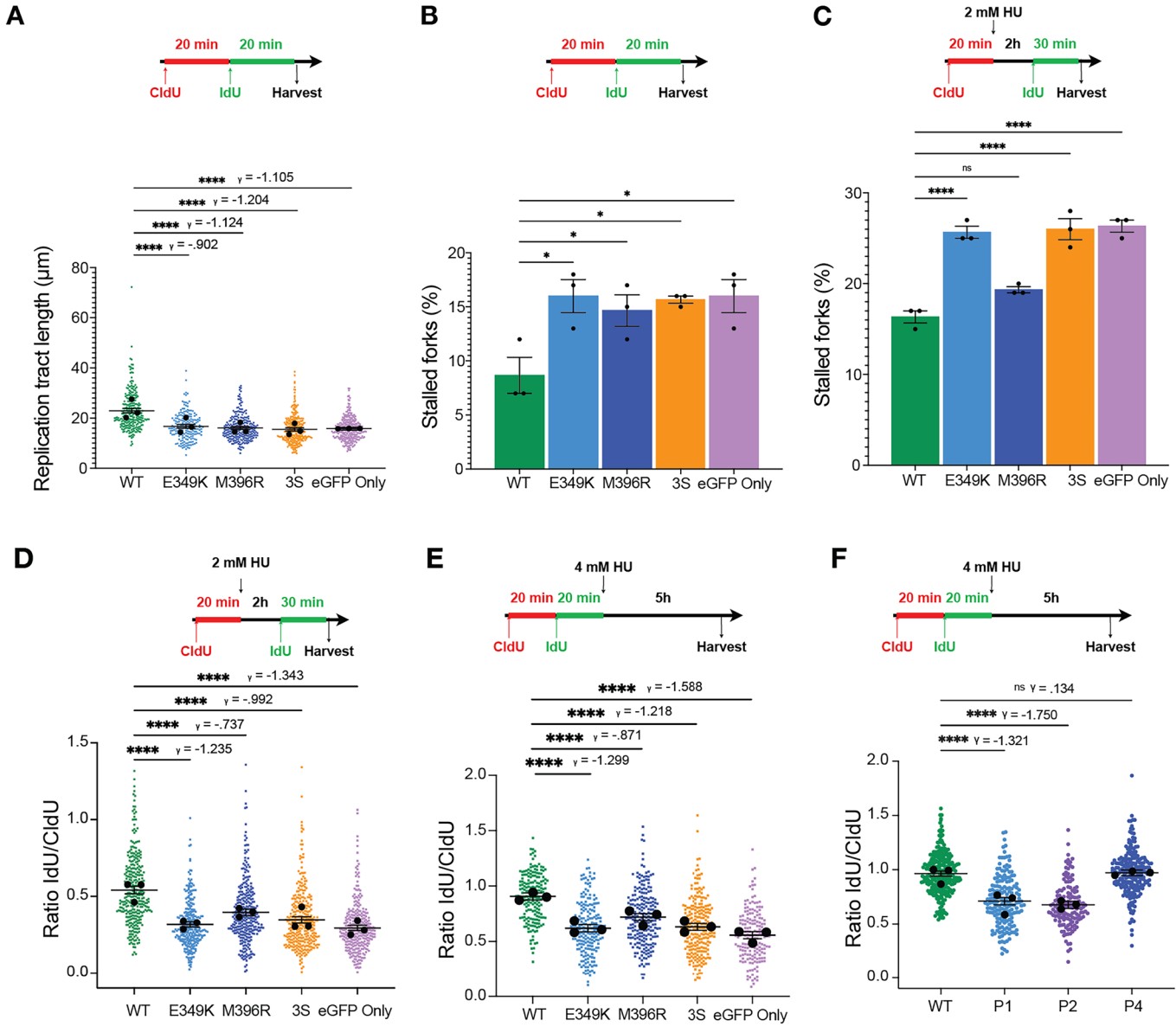

**Figure 5. MAEA loss impairs replication fork stability, restart, and protection.**

(A, B) Quantification of replication tract lengths (A) and spontaneous replication fork stalling (B) in *MAEA* KO U2OS cells complemented with the indicated eGFP expression constructs. (C, D) Quantification of HU-induced replication fork stalling (C) and fork restart after HU wash-out (D) in *MAEA* KO U2OS cells complemented with the indicated eGFP expression constructs. (E) Quantification of replication fork degradation after HU treatment in *MAEA* KO U2OS cells complemented with the indicated eGFP expression constructs. (F) Quantification of replication fork degradation in primary fibroblasts from Patients 1, 2, and 4 versus a WT control fibroblast cell line following treatment with HU. The data were the combined result of three independent experiments. *P* values were generated by performing a two-tailed Kruskal–Wallis test on scatter plots and an ordinary one-way ANOVA on histograms. γ is a measure of effect size. Bars represent the mean ± 95% CI. *$P \leq 0.05$, **$P \leq 0.01$, ***$P \leq 0.001$, ****$P \leq 0.0001$; exact *P* values can be found in Appendix Table S1. HU hydroxyurea, KO knockout. Source data are available online for this figure.

observed a severe replication fork progression defect in the presence of low-dose camptothecin in eGFP-only (*MAEA* KO) cells compared with eGFP-MAEA^WT (Fig. 4G). This indicated that MAEA loss compromises the ability of cells to resolve replication-stalling TOP1-associated DNA lesions.

We next assessed replication fork dynamics in the absence of exogenous replication stress and observed that *MAEA* KO cells expressing eGFP-only, eGFP-MAEA^3S, or two patient-associated variants, eGFP-MAEA^E349K (P1 and P7) and eGFP-MAEA^M396R (P4),

exhibited significantly reduced replication fork progression associated with increased spontaneous replication fork stalling (Fig. 5A,B). These data indicated that the E3 ubiquitin ligase activity of the CTLH complex is required to maintain faithful DNA replication in unperturbed conditions.

To assess whether the observed replication fork defects were further impacted by exogenous replication stress, we treated cells with short-term exposure to low millimolar doses of HU, inducing global and synchronous arrest of replication forks that can restart

following HU removal (Zellweger et al, 2015; Bhat and Cortez, 2018; Liu et al, 2023). After conducting DNA fiber assays, we observed that *MAEA* KO cells complemented with eGFP-only or eGFP-MAEA³ˢ exhibited increased fork stalling upon HU treatment and reduced efficiency of fork restart after HU wash-out compared with cells re-expressing eGFP-MAEA^WT (Fig. 5C,D). This indicated that the E3 ubiquitin ligase function of MAEA is required to properly restart transiently stalled replication forks.

We further observed that while cells expressing eGFP-MAEA^E349K exhibited comparable HU-induced replication fork stalling and restart efficiency to *MAEA* KO cells, eGFP-MAEA^M396R cells exhibited partially reduced replication fork restart efficiency without any apparent increase in replication fork stalling (Fig. 5C,D). These observations indicated that while MAEA^E349K is likely a null allele, MAEA^M396R is a hypomorphic allele that has differential impacts on replication fork stability and restart. In summary, these data indicated that *MAEA* KO and variant cell lines struggle with replication fork progression and restart.

RAD51 protects replication forks from nucleolytic degradation during replication fork reversal and aids their remodeling and/or restart (Bhat and Cortez, 2018; Qiu et al, 2021). Therefore, we considered that increased fork degradation might explain inefficient fork restart in *MAEA* KO cells. Thus, we incubated the panel of *MAEA* KO cells complemented with MAEA variants with CldU and IdU for 20 min each before a 5 h treatment with 4 mM HU to inhibit further DNA replication. We found that MAEA loss or mutation resulted in a reduced IdU:CldU ratio, indicative of increased fork degradation (Fig. 5E). However, *MAEA* KO cells expressing eGFP-MAEA^M396R displayed an intermediate replication fork degradation phenotype, consistent with our other findings assessing replication fork restart (Fig. 5C,D), suggesting that this mutation is hypomorphic.

Finally, we established immortalized skin fibroblasts from *MAEA* variant patients based on availability (P1, P2, and P4). Despite still retaining one WT *MAEA* allele, in DNA fiber assays, P1 (p.E349K) and P2 (splice-site mutation prior to the RING domain) cells exhibited significantly enhanced markers of fork degradation after exposure to high-dose HU compared with a non-isogenic WT control cell line. In contrast, cells from MAEA patient P4 (p.M396R) did not display a fork degradation phenotype (Fig. 5F). Given that our data modeling this mutation in *MAEA* KO cells indicated that this variant is hypomorphic on a null background (Fig. 5B–E), we surmise that in the presence of one WT *MAEA* allele, the dominant-negative impact of the mutant allele is tempered.

## Discussion

We have discovered that defects in MAEA and other components of the CTLH ubiquitin E3 ligase complex impair DNA replication and HR, causing hypersensitivity to anti-cancer agents such as hydroxyurea, the PARP inhibitor olaparib, and camptothecin (a close analog of clinically-used TOP1 poisons). Furthermore, we have identified apparent loss-of-function *MAEA* variants in eight individuals with the nonsyndromic DD/ID that we have termed DIADEM (**D**evelopmental delay and **I**ntellectual disability **A**ssociated with **DE**fects in **M**AEA). All eight DIADEM patients presented with developmental delay, intellectual disability, and delayed acquisition of speech. We also observed additional features, including brain abnormalities, reduced muscle tone and unsteady gait, immunodeficiency, and bone marrow failure with incomplete penetrance. Across the eight identified patients, six acquired different de novo variants with dominant effects, while the remaining two are siblings (P5a and P5b) who have the same homozygous variant, making it likely that this variant was inherited in a recessive manner (Tables EV2 and 3).

MAEA is a key subunit of the CTLH E3 ubiquitin ligase complex known to function in development (Goto and Shibuya, 2022; Briney et al, 2025). Our findings indicate that several *MAEA* variants likely cause DIADEM by compromising E3 ubiquitin ligase activity. Loss-of-function variants in another CTLH component (WDR26) cause SD syndrome, which shares some clinical features with DIADEM. However, there are notable differences, including the absence of seizures in five of the patients in our cohort. This suggests that DIADEM is distinct from SD syndrome. We speculate that variants in other CTLH components may yield overlapping but non-identical conditions due to their closely connected but non-identical functions.

Our data indicate that cells lacking MAEA are HR deficient, and that while they are proficient in DNA end resection, RAD51 loading onto ssDNA is impaired, which could explain the hypersensitivity of these cells towards agents that induce seDSBs. Various core HR factors, including RAD51, BRCA1, and BRCA2, are associated not only with DSB repair but also with cellular tolerance of DNA replication stress (Kolinjivadi et al, 2017). Likewise, MAEA promotes both effective HR and faithful DNA replication. Accordingly, we also found that, in addition to exhibiting increased DSBs and survival defects upon exposure to replication stress-inducing agents, MAEA-deficient cells experience more replication fork stalling under these conditions and less efficient fork restart after release from hydroxyurea.

We determined that the requirement for MAEA in HR and replication stress depends on its E3 ubiquitin ligase activity. In line with our in silico analysis of MAEA DIADEM variants, cell models of the patient variants E349K (P1/P7) and M396R (P4) phenocopied those cells expressing the 3S RING domain mutant, suggesting that DIADEM can be caused by loss of CTLH ubiquitin ligase activity. However, we note that immortalized fibroblasts from patient P4 did not exhibit detectable replication fork protection defects upon hydroxyurea treatment, in contrast to cells from patients P1 or P2. Indeed, even in isogenic U2OS cell lines, P4's M396R variant consistently exhibited milder phenotypes in assays related to replication stress compared with WT MAEA, despite behaving similarly in viability assays under camptothecin treatment. We speculate that this variant, which is outside of the MAEA RING domain, may have some residual function. The range of variants and the large number of known CTLH substrates (Maitland et al, 2021, 2022) suggest that the etiology of DIADEM is complex and that its presentation could depend upon the nature of the MAEA variant, as well as the developmental stage of the individual.

While this manuscript was in preparation, another study reported MAEA-mediated polyubiquitylation and degradation of PARP1 in models of gastric cancer and colorectal cancer, with MAEA overexpression conferring cellular sensitivity to oxaliplatin (Feng et al, 2025). Since oxaliplatin induces DNA crosslinks that require HR for their repair (Semlow and Walter, 2021), we consider

it unlikely that a role for MAEA in regulating PARP1 underlies the breadth of HR- and DNA replication-related findings described in our study. Moreover, the Feng et al study suggests that MAEA loss leads to veliparib (PARP inhibitor) resistance and that, conversely, overexpression generates sensitivity. This is inconsistent with our own findings that loss of MAEA and CTLH components sensitizes cells to PARP inhibition and may be attributable to the cell model used in the Feng et al study. A further study published while this manuscript was in revision reported findings that align well with our own conclusions: that MAEA loss compromises RAD51 loading at DNA damage sites and leads to hypersensitivity to replication stress- and seDSB-inducing agents. Mechanistically, Zeinali et al propose that MAEA polyubiquitylates KU80 to promote its removal from DNA ends (Zeinali et al, 2025). It will be clinically important to explore the extent to which this function may impact DIADEM patients.

In our study, MAEA-deficient cells exhibit phenotypes reminiscent of clinical HR deficiency (HRD) as observed in cancer cells. Impaired RAD51 foci formation is a well-established biomarker for HRD and PARPi efficacy in the clinic, making loss or deregulation of the CTLH complex potential prognostic biomarkers for PARPi treatments. Several CTLH components have been found in CRISPR screens as candidate mediators of PARP inhibition and replication stress (DeWeirdt et al, 2020; Benslimane et al, 2020; Tessari et al, 2020; Jamal et al, 2022), but until now, these phenotypes have remained unexplored. We note that loss of expression of CTLH components *MAEA* and *YPEL5* have been reported in subsets of tumor types (e.g., breast and ovarian) for which PARPi therapies are approved (Jamal et al, 2022) which highlights the potential for CTLH components as prognostic biomarkers.

Unlike *BRCA1* or *BRCA2* deficiency, DIADEM does not currently appear to predispose to cancer. However, due to the young age of the patients in our cohort, the impact of MAEA variants on aging-associated diseases is unknown; long-term monitoring of these patients will be an important aspect of their care. Moreover, based on our findings, we caution that should a patient with DIADEM develop cancer, the use of chemotherapies that are highly effective in HRD cancers (platinum agents, olaparib, and derivatives of camptothecin such as irinotecan and topotecan) might lead to severe toxicity. These considerations were not the focus of our study, but might be relevant to patients with pathological variants in other CTLH complex members, including SD syndrome patients.

# Methods

### Reagents and tools table

| Reagent/resource | Reference or source | Identifier or catalog number |
| --- | --- | --- |
| **Experimental models** | | |
| U2OS-Cas9 | This study/ATCC derivative | N/A |
| HAP1 | Horizon Discovery | C631 |
| 293FT | Thermo Fisher Scientific | R70007 |

| Reagent/resource | Reference or source | Identifier or catalog number |
| --- | --- | --- |
| Primary dermal fibroblasts | This study | N/A |
| hTERT-immortalized fibroblasts | This study | N/A |
| SH-SY5Y | Gift from Kevin Brindle | N/A |
| **Recombinant DNA** | | |
| pKLV2-U6gRNA5(BbsI)-PGKpuro2ABFP-W | Addgene | #67974 |
| psPax2 | Addgene | #12260 |
| pMD2.G | Addgene | #12259 |
| pLV-hTERT-IRES-hygro | Addgene | #85140 |
| lentiCRISPR-v2 | Addgene | #52961 |
| **Antibodies** | | |
| GFP | Invitrogen | A11122 |
| LAMIN B1 | Abcam | ab16048 |
| MAEA | R&D | AF7288 |
| RMND5A | Novus | NBP1-92337 |
| RAD51 | Santa Cruz | sc-8349 |
| TOP1 | TopoGEN | TG2012 |
| VINCULIN | Abcam | ab219649 |
| EXO1 | Proteintech Europe Ltd | 16253-1-AP |
| CtIP | Fisher Scientific UK Ltd | MA1-23304 |
| ATM | Abcam | ab32420 |
| BRCA2 | Calbiochem/Merck Biosciences Ltd | OP95 |
| BRCA1 | MilliPoreSigma | 07-434 |
| DNA2 | Proteintech Europe Ltd | 18727-1-AP |
| β-ACTIN | Cell Signaling Technology | 4970S |
| β-ACTIN | Abcam | ab8226 |
| γH2AX (S139) | Cell Signalling Technology | 2577 |
| BrdU | GE Healthcare | RPN20AB |
| CENPF | Abcam | ab223847 |
| PCNA | Abcam | ab18197 |
| γH2AX (S139) | Millipore | 05-636 |
| RAD51 | Abcam | ab88572 |
| RPA | Lab Vision | MS-691-P1 |
| Goat Anti-Rabbit Alexa Fluor 488 | Invitrogen | A11034 |
| Goat Anti-Mouse Alexa Fluor 488 | Invitrogen | A11029 |
| Goat Anti-Rabbit Alexa Fluor 594 | Invitrogen | A11037 |
| Goat Anti-Mouse Alexa Fluor 594 | Invitrogen | A11032 |

| Reagent/resource | Reference or source | Identifier or catalog number |
|---|---|---|
| Goat Anti-Rabbit Alexa Fluor 647 | Invitrogen | A21245 |
| Goat Anti-Mouse Alexa Fluor 647 | Cambridge Bioscience | CUK0357 C04-64 |
| Rabbit Anti-Mouse | Dako Ltd | P0260 |
| Goat Anti-Mouse | Millipore | AP124P |
| Goat Anti-Rabbit | Perbio | 31462 |
| Rabbit Anti-Sheep | Dako Ltd | P0163 |
| Rabbit Anti-Goat | Dako Ltd | P0449 |
| IRDye 800CW Donkey Anti-Mouse | LI-COR BIOSCIENCES UK Ltd | 926-32212 |
| IRDye 680LT Goat Anti-Rabbit | LI-COR BIOSCIENCES UK Ltd | 926-68021 |
| **Oligonucleotides and other sequence-based reagents** | | |
| sgMAEA_1_F (Cloning Guide) | This study | |
| sgMAEA_1_R (Cloning Guide) | This study | |
| MAEA_1_F (Primer) | This study | |
| MAEA_1_R (Primer) | This study | |
| **Chemicals, enzymes and other reagents** | | |
| PVDF membrane | Amersham | GE10600023 |
| Nitrocellulose membrane | Amersham | GE10600002 |
| Dulbecco's Modified Eagle Medium | Thermo Fisher Scientific | 11965092 |
| DMEM/F-12 medium | Thermo Fisher Scientific | 11320033 |
| Iscove's Modified Dulbecco's Medium | Thermo Fisher Scientific | 12440053 |
| Penicillin-Streptomycin-Glutamine | Thermo Fisher Scientific | 10378016 |
| Fetal Bovine Serum (FBS) | Thermo Fisher Scientific | 10500064 |
| Blasticidin (BSD) | Thermo Fisher Scientific | A1113903 |
| Puromycin dihydrochloride | Merck/Sigma-Aldrich | P8833 |
| Hygromycin B | Thermo Fisher Scientific | 10687010 |
| Trypsin-EDTA (0.25%) | Gibco | 25200056 |
| Crystal Violet | Merck | C0775 |
| Camptothecin | Merck | C9911 |
| Olaparib | Selleck Chemicals | S1060 |
| Etoposide | Merck | E1383 |
| HEPES | Merck | H3375 |
| Triton X-100 | Merck | X-100 |
| Sucrose | Merck | S0389 |
| Paraformaldehyde (PFA) | Electron Microscopy Sciences | 15710 |

| Reagent/resource | Reference or source | Identifier or catalog number |
|---|---|---|
| Tween 20 | Merck | P1379 |
| Bovine Serum Albumin (BSA) | Merck | A9647 |
| DAPI | Thermo Fisher Scientific | D1306 |
| CldU (5-chloro-2′-deoxyuridine) | Merck | C6891 |
| IdU (5-iodo-2′-deoxyuridine) | Merck | I7125 |
| Hydroxyurea (HU) | Merck | H8627 |
| **Software** | | |
| DrugZ | Colic et al, 2019 | https://github.com/hart-lab/drugz |
| GraphPad Prism v10.01 | GraphPad Software | RRID:SCR_002798 |
| Harmony v5.1 | PerkinElmer | RRID:SCR_023543 |
| FlowJo | BD Biosciences | RRID:SCR_008520 |
| FIJI (ImageJ) | N/A | RRID:SCR_002285 |
| PyMOL v3.0 | Schrödinger, LLC | RRID:SCR_000305 |
| Perfolizer | Akinshin (GitHub, 2023) | https://github.com/AndreyAkinshin/perfolizer |
| **Other** | | |
| ChemiDoc MP Imaging System | Bio-Rad | 12003154 |
| Opera Phenix Plus High-Content Microscope | PerkinElmer | RRID:SCR_021100 |
| Beckman CytoFLEX LX | Beckman Coulter | N/A |
| Thermo Fisher Attune NxT | Thermo Fisher Scientific | A24858 |
| X-ray film processor | Generic | N/A |
| 24-well sensoplate (IF) | Greiner BIO-ONE | 662892 |

## Study approval

Written informed consent to publish clinical information of the affected individuals was obtained from the families prior to their involvement in this study, in accordance with local IRB-approved protocols. Further approval for this research was obtained from the West Midlands, Coventry, and Warwickshire Research Ethics Committee (Coventry, United Kingdom; REC: 20/WM/0098). All experiments conformed to the principles set out in the WMA Declaration of Helsinki and the Department of Health and Human Services Belmont Report.

## CRISPR-Cas9 screen

A custom library of CRISPR guides (Table EV1) targeting 886 E3 ligases and related proteins was cloned into pKLV2-U6gRNA5(BbsI)-PGKpuro2ABFP-W (Addgene #67974), packaged into lentiviral particles using second-generation plasmids psPax2 (Addgene #12260) and pMD2.G (Addgene #12259), and titered. Three independent populations of WT U2OS-Cas9 cells were

transduced at a multiplicity of infection of 0.25 at 500X representation and selected using puromycin (2 μg/mL) for 14 days. Cells were subjected to $IC_{50}$ camptothecin (12 nM, 9 days). Cells were allowed to recover for three days with no treatment. DNA from pre- and post-treatment pooled cell populations were extracted and PCR amplified with relevant next-generation sequencing barcodes. Sequencing results were analyzed by using DrugZ (Colic et al, 2019).

## Cell culture and generation of cell lines

U2OS Cas9 cells were transfected with guides (IDT) targeting MAEA, designed using Guide Picker and CRISPOR (Table EV4) (Hough et al, 2017; Haeussler et al, 2016). After transfection with RNAiMAX (Thermo Fisher Scientific), cells were grown as single colonies. Genomic DNA extraction, PCR, Sanger sequencing, and TIDE analysis (Brinkman et al, 2014) were used to identify KO clones. Dermal primary fibroblasts were grown from skin-punch biopsies and maintained in DMEM supplemented with 20% FCS, 5% L-glutamine, and 5% penicillin-streptomycin antibiotics (Merck). Primary fibroblasts were immortalized by lentiviral transduction with hTERT that was generated by transfecting 293FT cells (Thermo Fisher Scientific) with pLV-hTERT-IRES-hygro (Addgene #85140). Cells were selected with hygromycin (Thermo Fisher Scientific) at 70 μg/mL. U2OS cells were grown in DMEM plus 1X PSQ (Thermo #10378016) and 10% FBS. SH-SY5Y cells were grown in DMEM/F-12 medium plus 1X PSQ and 10% FBS. HAP1 cells were grown in IMDM with 1X PSQ and 10% FBS. All cell lines were validated mycoplasma-free and have not undergone recent STR authentication.

## hTERT fibroblast immortalization

Dermal primary fibroblasts were grown from skin-punch biopsies and maintained in DMEM (Thermo Fisher Scientific) supplemented with 20% FCS, 5% L-glutamine and 5% penicillin-streptomycin. Primary fibroblasts were immortalized with a lentivirus expressing human telomerase reverse transcriptase (hTERT) that was generated by transfecting 293FT cells (Thermo Fisher Scientific) with the plasmids: pLV-hTERT-IRES-hygro (Addgene #85140), psPax2 (Addgene #12260) and pMD2.G (Addgene #12259). Selection was performed using hygromycin (Thermo Fisher Scientific) at 70 μg/mL.

## Clonogenic survival assays

6-well plates were seeded with 500 U2OS or 250 HAP1 cells in technical triplicate. After 8 h or overnight of attaching/growth following initial seeding, all medium was replaced with medium containing the appropriate drug dilution. After 10–14 days, cells were washed with PBS and stained with crystal violet. Colonies were analyzed after scanning plates and using FIJI. DNA-damaging agents: olaparib (0.5–2 μM U2OS), etoposide (25–100 nM U2OS), camptothecin (2.5–7 nM U2OS). Data shown represent a minimum of three independent experiments. Each experiment consists of the averaged values from three technical replicates. To analyze seeding density, 250, 500, 750, and 1000 U2OS cells were seeded per well and analyzed as above.

## Immunoblotting

Following SDS-PAGE and wet transfer, PVDF (Amersham Hybond P0.45) or nitrocellulose (Amersham Protran 0.45) membranes were blocked in 5% BSA/TBST for 1 h at room temperature on a rocker. Antibodies were diluted in 1% BSA/TBST as detailed in Table EV5. Membranes were incubated with the primary antibody, washed three times, and incubated with the secondary antibody. Signals were detected using chemiluminescence, an X-ray film processor, or using a ChemiDoc MP Imaging System (RRID:SCR_019037, Bio-Rad).

## Immunofluorescence

SH-SY5Y were transduced with lentiCRISPR-v2 (Addgene #52961) containing sgMAEA1.0 (Table EV2) and selected in puromycin (2 μg/ml). Cells were seeded in 24-well glass-bottom plates with or without BrdU (Merck, B9285). After 24 h (U2OS) and after 48 h (SH-SY5Y), cells were treated with CPT (1 μM, 1 h) or DMSO. Cells were washed and pre-extracted with ice-cold CSK (25 mM HEPES pH 7.4, 50 mM NaCl, 3 mM $MgCl_2$, 0.5% Triton X-100, and 300 mM sucrose) on ice (3 min) prior to fixation in 2% PFA (10 min). Cells were washed with PBS-0.2% Tween 20 (PBST), blocked in 5% BSA and stained with primary antibodies in 1% BSA. Antibody details can be found in Table EV6. After primary antibody incubation, wells were washed three times with PBST and then incubated with secondary antibody, 2 μg/mL DAPI. Images were acquired using an Opera Phenix Plus High-Content microscope and analyzed with Harmony v.5.1. Cells were stratified by cell cycle markers (PCNA/CENPF or γH2AX) to identify S-phase nuclei.

## Traffic light reporter (TLR) assay

TLR assays were conducted as described previously (11). Cells were counted on either a Beckman CytoFLEX LX or Thermo Fisher Attune NxT (consistent within the set of replicates). Data were then analyzed on FlowJo. Gates were set with an internal negative control population expressing neither BFP nor IFP. Background cell counts were subtracted from all totals. Data represent at least three independent experiments. siLuciferase, a negative control, and siCtIP, targeting a known HR factor, were performed 6 times. siRNA details can be found in Table EV7.

## Statistical tests

The unpaired *t*-test, two-tailed Kruskal–Wallis test, and ordinary one-way analysis of variance (ANOVA) tests were generated using GraphPad Prism v.10.01. Effect size (γ) using the C# GitHub repository Perfolizer by Andrey Akinshin (https://github.com/AndreyAkinshin/perfolizer) (Akinshin, 2023). γ is similar to a non-parametric Cohen's d and is a measure of the pooled median absolute deviations that fit between the median of group X (e.g., *MAEA* WT) and the median of group Y (e.g., *MAEA* KO). In general, the absolute value of γ can be evaluated according to the guidelines 0.00–0.10 (Negligible), 0.10–0.20 (Weak), 0.20–0.40 (Moderate), 0.40–0.60 (Moderately Strong), 0.60–0.80 (Strong), and 0.80 < (Very Strong).

**The paper explained**

**Problem**

DNA repair and DNA replication are intimately tied to diseases like cancer and neurodevelopmental conditions. The full landscape of DNA damage response (DDR) genes is not yet known, leading to gaps in clinical understanding. Eight patients with variants in the gene *MAEA*, part of the large CTLH E3 ubiquitin ligase complex, present with global developmental delay and other craniofacial differences. To date, the genetic basis for this condition remains undescribed.

**Results**

MAEA null cells are highly vulnerable to chemotherapeutics like the PARP inhibitor olaparib. Loss of MAEA impairs cellular ability to load RAD51, a recombinase that promotes replication fork protection and restart and drives DNA repair by means of homologous recombination (HR). Variants from the DIADEM patient cohort similarly diminish the DNA repair and replication capacity in both model cell systems and patient fibroblasts.

**Impact**

Uncovering a novel regulator of HR and DNA replication provides an untapped therapeutic axis for stratifying patients and treating a range of diseases like cancer. Moreover, describing the clinical presentation of a cohort of DIADEM patients and offering a genetic basis for the condition could provide early diagnostic criteria to help clinicians and families intervene and improve patients' quality of life.

## DNA fiber assay

Cells were pulse-labeled with 25 µM CldU for 20 min, washed with PBS, pulse-labeled for 20 min with 250 µM IdU, and then harvested. For replication restart experiments, cells were labeled with CldU for 20 min, washed in warm PBS, and incubated in medium containing 2 mM HU for 2 h. Cells were washed again in warm PBS and then incubated with 250 µM IdU for 20 min. Replication in the presence of replication stress was assayed by first pulse labeling cells with 25 µM CldU, which was washed off with medium containing 250 µM IdU and 100 nM CPT. Cells were then pulse labeled with 250 µM IdU and 100 nM CPT, for untreated cells CPT was omitted, at the end of the pulse labeling the cells were harvested as previously described. For fork protection experiments, cells were labeled with 25 µM CldU for 20 min, washed with CldU-containing medium, labeled with 250 µM IdU for 20 min, washed with warmed medium containing 4 mM HU, and incubated in medium containing HU for 5 h. DNA fiber analysis was carried out as previously described (Higgs et al, 2015).

## Structural analysis

The potential impact of the MAEA mutations was assessed by inspection of the molecular context of each variant residue in the MAEA/RMND5A/UBE2H complex crystal structure (PDBID: 8PJN (Chrustowicz et al, 2024)) using PyMOL (PyMOL molecular graphics system, V 3.0, Schrodinger LLC). Missing side chains were modeled manually by selecting rotamers with the lowest clash score.

## Experimental design

No blinding was done for the purposes of these studies.

## Data availability

This study includes no data deposited in external repositories.

The source data of this paper are collected in the following database record: biostudies:S-SCDT-10_1038-S44321-025-00352-x.

## Peer review information

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

## Acknowledgements

We thank the patients and parents for taking part in this study. We thank M. Agudo for help with the CRISPR screen. We acknowledge K. Harnish of the Gurdon Institute for performing next-generation sequencing on our CRISPR-Cas9 screening samples and the Gurdon and CRUK Cambridge Institute core facilities for assistance and support. Research in the SPJ laboratory is supported by Cancer Research UK (CRUK) Discovery Award DRCPGM \100005, CRUK Cambridge Institute core grant SEBINT-2024/100003 and ERC Synergy Award 855741 (DDREAMM). SHH was supported by Wellcome Investigator Award 206388/Z/17/Z and CRUK Discovery Award DRCPGM \100005; SWA by a Mark Foundation for Cancer Research (MFCR) ASPIRE II Award; is a recipient of the Women's Postdoctoral Career Development Award in Science from the Weizmann Institute of Science; and was a recipient of an Outstanding Postdoctoral Women Fellowship from the Israeli Council for Higher Education; SL by ERC Synergy Award 855741; JCT by Wellcome Investigator Award 206388/Z/17/Z and ERC Synergy Award 855741; CJC, GZV, RB, and YG by CRUK Discovery Award DRCPGM\100005; ASB by CRUK RadNET Cambridge C17918/A28870 and Wellcome Early Career Award 227014/Z/23/Z; and OL by CRUK Cambridge Institute core grants C9545/ A29580 and SEBINT-2024/100003. The SPJ laboratory was also supported by CRUK Program grant C6/A18796 and core funding grants C6946/A24843 and WT203144 to the Gurdon Institute. GSS and SSJ are funded by a CRUK Programme grant (C17183/A23303) and an MRC project grant (UKRI577). CJC is supported by a grant from the Deutsche Forschungsgemeinschaft (DFG; KU 563/18-1). Research in the Beli lab is funded by the DFG project-ID 393547839—SFB 1361. This research was made possible through access to data in the National Genomic Research Library, which is managed by Genomics England Limited (a wholly owned company of the Department of Health and Social Care). The National Genomic Research Library holds data provided by patients and collected by the NHS as part of their care, and data collected as part of their participation in research. The National Genomic Research Library is funded by the National Institute for Health Research and NHS England. The Wellcome Trust, Cancer Research UK and the Medical Research Council have also funded research infrastructure. For the purpose of open access, we have applied a Creative Commons Attribution (CC BY) public copyright licence to any Author Accepted Manuscript version arising from this submission.

## Author contributions

**Søren H Hough**: Conceptualization; Formal analysis; Investigation; Methodology; Writing—original draft; Writing—review and editing. **Satpal S Jhujh**: Conceptualization; Formal analysis; Investigation; Visualization; Writing—review and editing. **Samah W Awwad**: Formal analysis; Investigation; Visualization; Writing—review and editing. **Oliver E Lewis**: Formal analysis; Investigation; Visualization; Writing—review and editing. **Simon Lam**: Resources; Software; Formal analysis; Visualization. **John C Thomas**: Resources; Software; Formal analysis; Visualization. **Thorsten Mosler**: Investigation; Writing—review and editing. **Aldo Bader**: Conceptualization; Formal analysis; Visualization. **Lauren Bartik**: Conceptualization; Investigation; Project administration. **Shane McKee**: Formal analysis; Investigation; Project administration. **Shivarajan Amudhavalli**: Formal analysis; Investigation. **Estelle Colin**: Investigation. **Nadirah Damseh**: Investigation. **Emma Clement**: Investigation. **Pilar Cacheiro**: Investigation. **Anirban Majumdar**: Investigation. **Damian Smedley**: Investigation; Project administration. **Joël Fluss**: Investigation. **Rosalinda Gianinni**: Investigation; Writing—review and editing. **Isabelle Thiffault**: Project administration. **Guido Zagnoli Vieira**: Conceptualization; Supervision; Investigation. **Rimma Belotserkovskaya**: Conceptualization; Supervision; Writing—review and editing. **Stephen J Smerdon**: Formal analysis. **Petra Beli**: Supervision. **Yaron Galanty**: Conceptualization; Supervision; Methodology; Project administration. **Christopher Carnie**: Conceptualization; Formal analysis; Supervision; Validation; Investigation; Visualization; Writing—original draft; Project administration; Writing—review and editing. **Grant S Stewart**: Conceptualization; Formal analysis; Supervision; Funding acquisition; Project administration; Writing—review and editing. **Stephen P Jackson**: Supervision; Funding acquisition; Project administration; Writing—review and editing.

Source data underlying figure panels in this paper may have individual authorship assigned. Where available, figure panel/source data authorship is listed in the following database record: biostudies:S-SCDT-10_1038-S44321-025-00352-x.

## Disclosure and competing interests statement

SPJ and YG work part-time at Insmed Innovation UK Ltd. SPJ is a founding partner of Ahren Innovation Capital LLP, a co-founder of Mission Therapeutics Ltd, and is a consultant and shareholder of Genome Therapeutics Ltd. The authors declare no other competing interests.

# Expanded View Figures

**Figure EV1.  *MAEA* KO and HM cell line validation.**

(A) Immunoblot for MAEA in the indicated U2OS cell lines; $n = 2$ independent experiments. (B) TIDE analysis of the sgRNA target site in *MAEA* KO and *MAEA* HM cells. (C) Sanger trace analysis of the sgRNA target site in *MAEA* KO and HM cells. (D) Cell cycle analysis of *MAEA* KO and HM cells. (E, F) Representative images of untreated WT and MAEA-deficient U2OS (E) and HAP1 (F) cells. (G) Immunoblot for MAEA in U2OS following siRNA transfection; $n = 3$ independent experiments. (H, I) Representative images (H) and colony counts (I) of U2OS cells with or without siRNA-mediated MAEA depletion. (J) The average colony size of siRNA-transfected U2OS cells at 500 cells per well. (K) Total area coverage of U2OS colonies following siRNA transfection. (I–K) Bars represent the mean ± SEM. KO knockout, HM hypomorph. Source data are available online for this figure.

▶

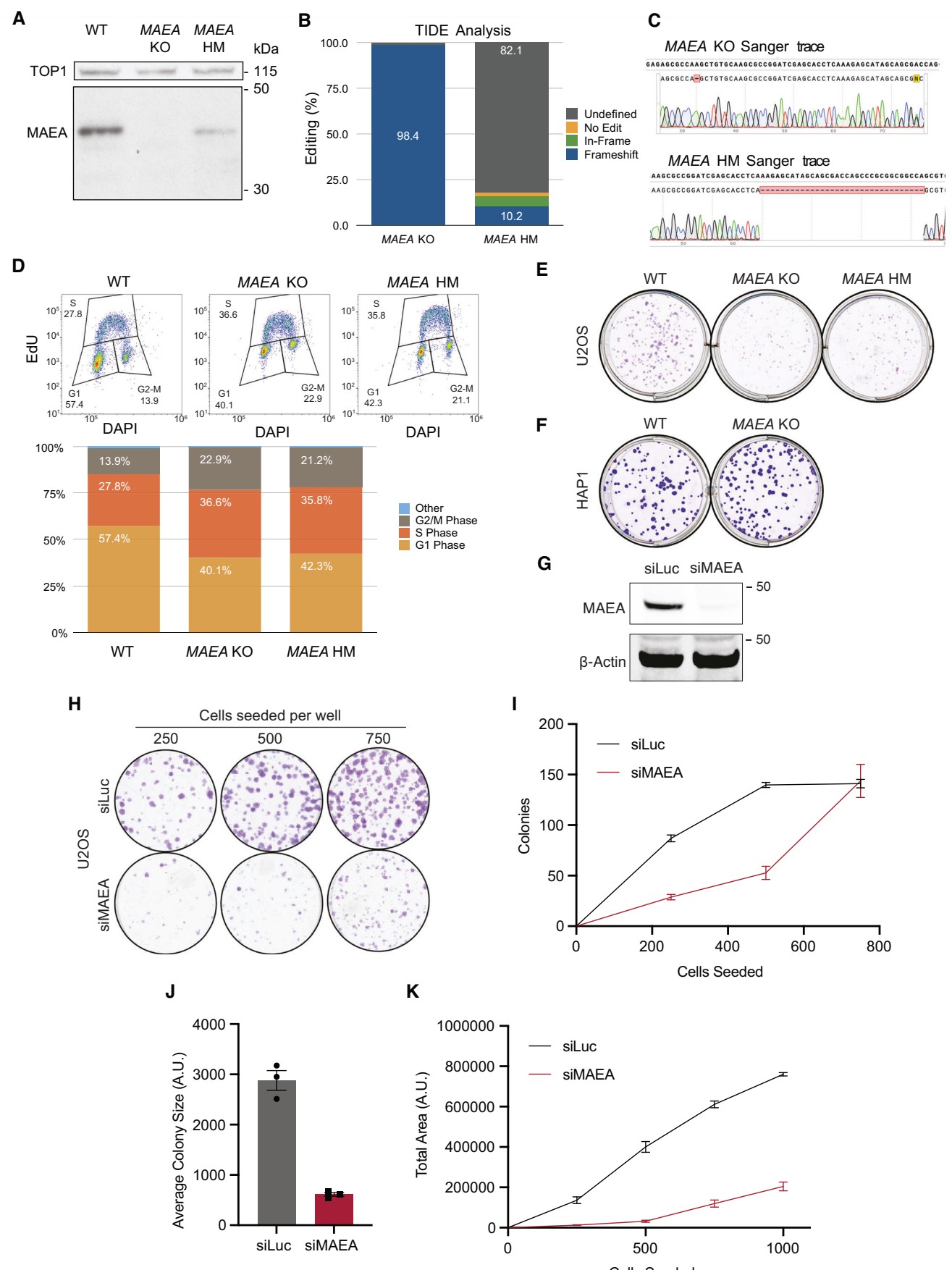

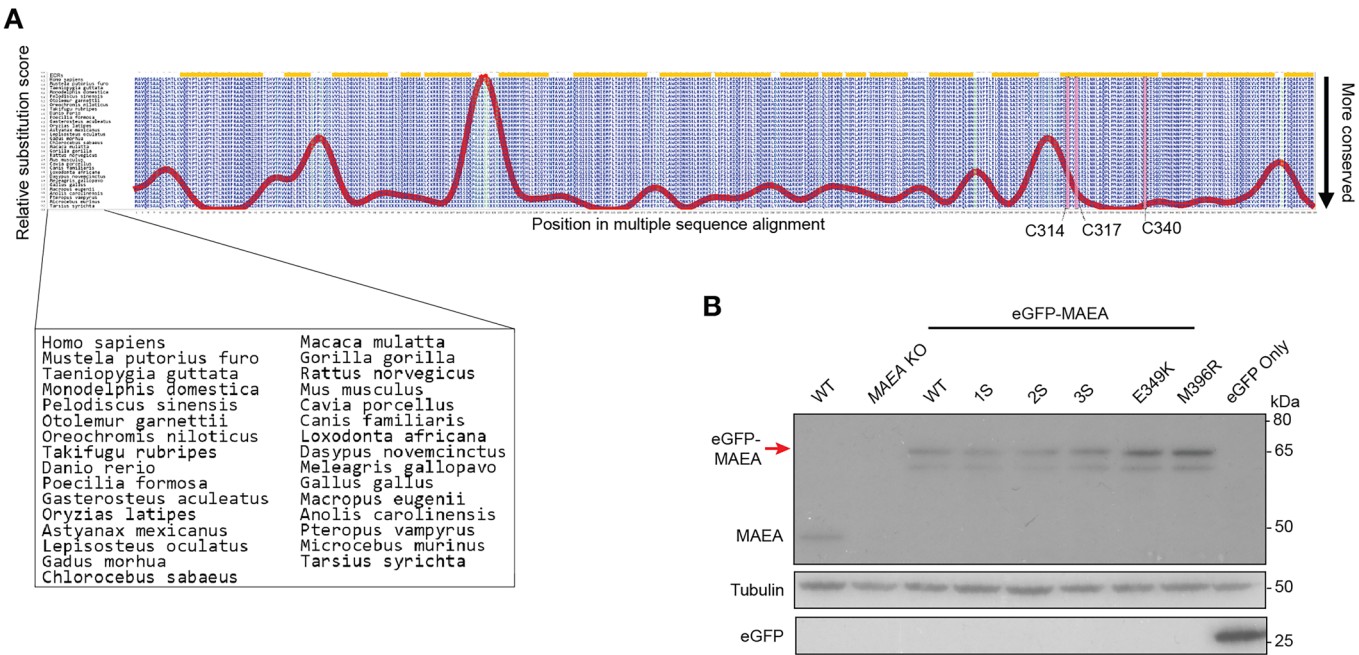

**Figure EV2. Clinical and C > S MAEA variants are evolutionarily conserved.**

(A) Aminode analysis (Chang et al, 2018) of MAEA, with C > S mutations annotated. The red line represents conservation across species. Species compared in the analysis are listed in the box. (B) Immunoblot assessing expression of eGFP-MAEA constructs; $n = 2$ independent experiments. Source data are available online for this figure.

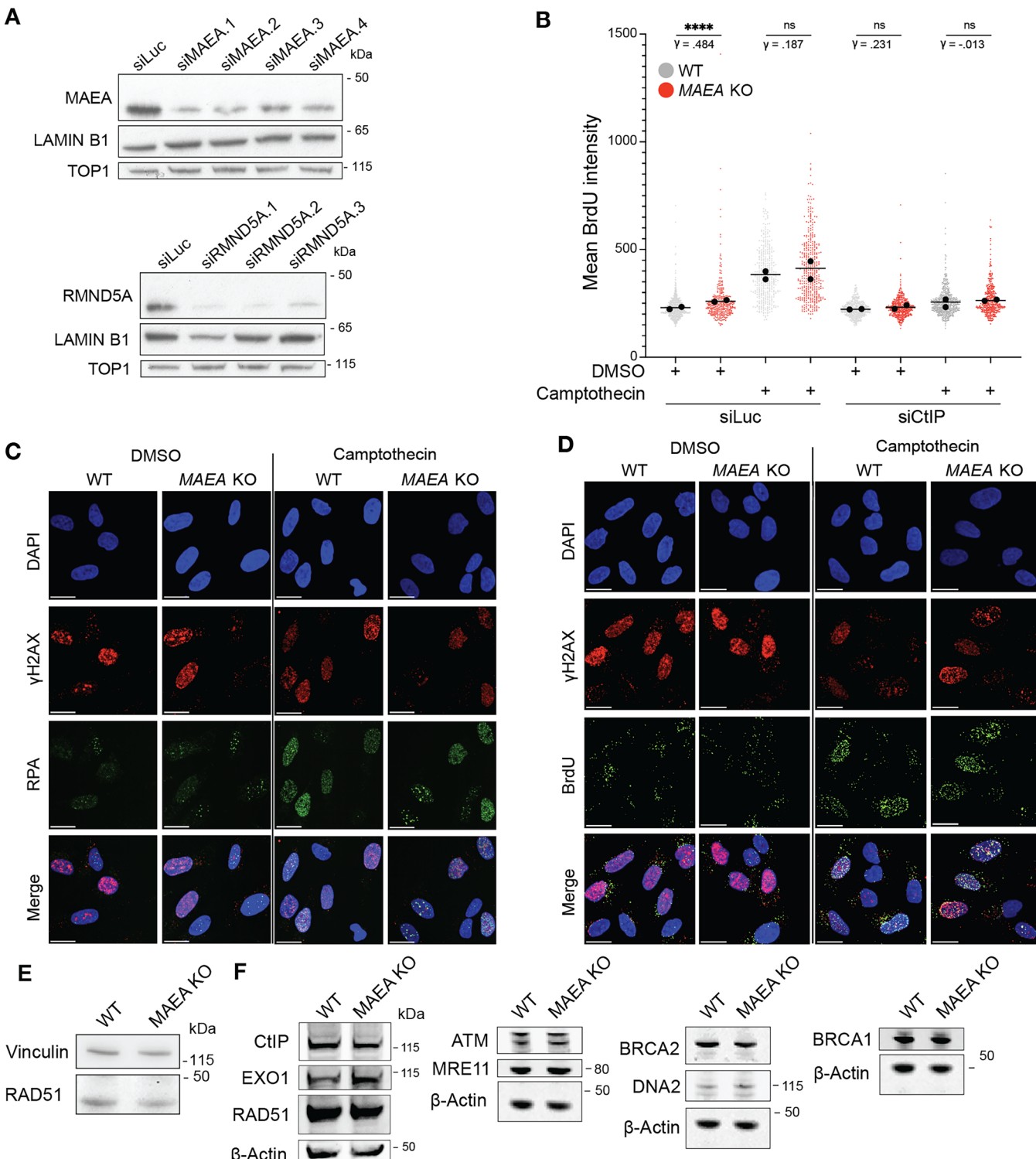

**Figure EV3. MAEA loss compromises RAD51 foci formation.**

(A) Immunoblot against the indicated proteins with the indicated siRNAs; $n = 3$ independent experiments. (B) Quantification of BrdU immunofluorescence, after camptothecin treatment in the presence or absence of siRNA-mediated CtIP depletion. (C, D) Representative images from Fig. 3C,E expanded to include γH2AX staining, indicative of the S phase cells. (E) Immunoblot against RAD51 in WT and *MAEA* KO U2OS cells. (F) Immunoblot for indicated proteins in U2OS WT or *MAEA* KO cells. (E, F) are each $n = 3$ independent experiments. *P* values were generated by performing a two-tailed Kruskal–Wallis test. γ is a measure of effect size. The data were the combined results of two independent experiments. Bars denote mean ± 95% CI. *$P \leq 0.05$, **$P \leq 0.01$, ***$P \leq 0.001$, ****$P \leq 0.0001$; exact *P* values can be found in Appendix Table S1. Scale bars = 20 μm. NT non-treated, KO knockout. Source data are available online for this figure.

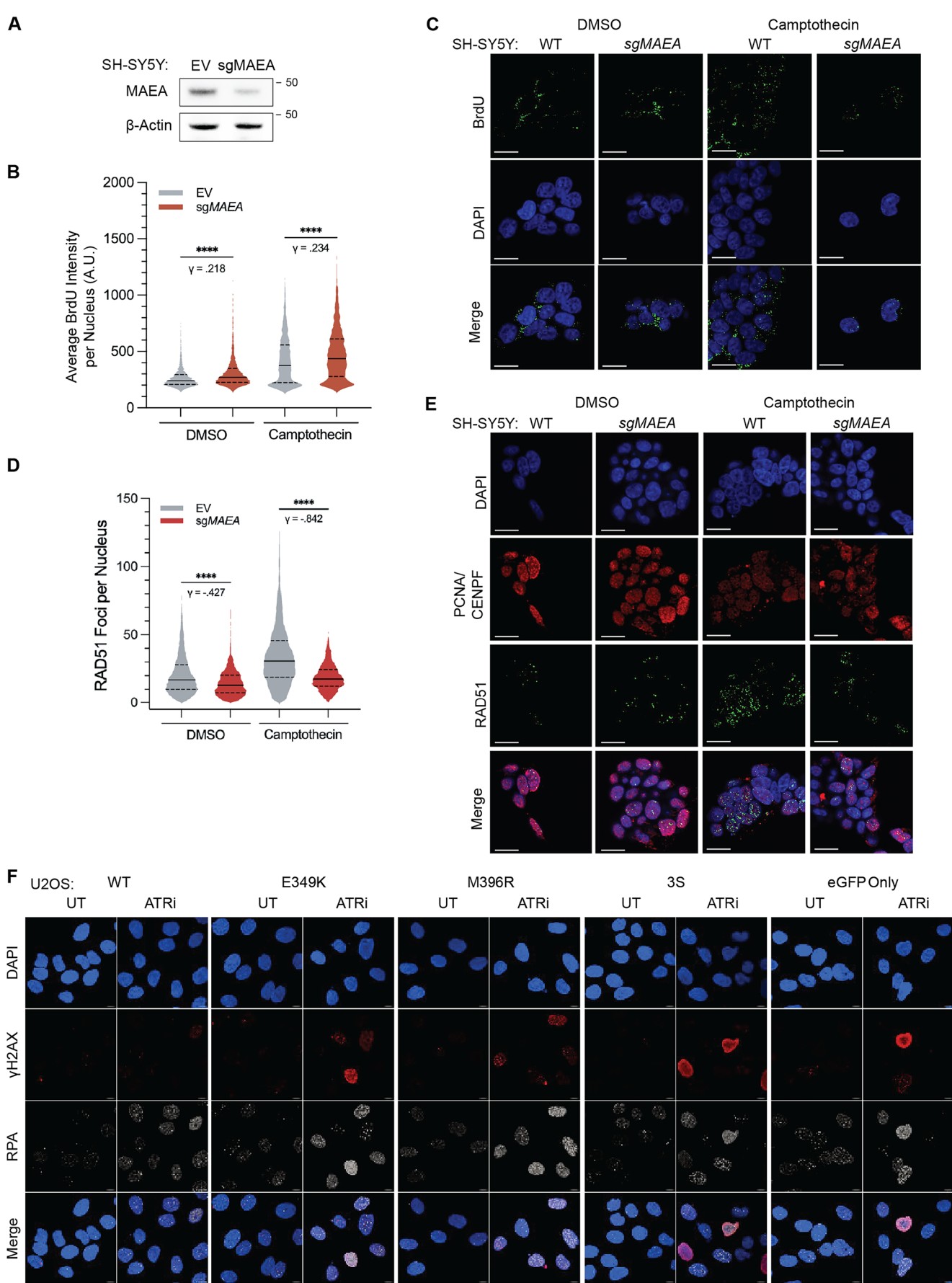

◄ **Figure EV4. RAD51 loading defects in MAEA-deficient SH-SY5Y cells.**

(A) Immunoblot for MAEA depletion in polyclonal SH-SY5Y cells following CRISPR-Cas9 editing; $n = 3$ independent experiments. (B) Quantification and (C) representative images of BrdU in SH-SY5Y cells treated with DMSO (1 h) or camptothecin (1 μM, 1 h). (D) Quantification and (E) representative images of RAD51 foci in S-phase SH-SY5Y cells treated with DMSO or camptothecin. Bars in (B, D) represent median and interquartile range. *P* values were generated by performing a two-tailed Kruskal–Wallis test. γ is a measure of effect size. The data represent three independent experiments. *$P \leq 0.05$, **$P \leq 0.01$, ***$P \leq 0.001$, ****$P \leq 0.0001$; exact *P* values can be found in Appendix Table S1. Scale bars = 10 μm. For (B, C), outliers were removed using ROUT analysis (Q = 1%). (F) Representative images of Fig. 5F. Source data are available online for this figure.

