## [Peer Review File · EMBO Molecular Medicine]

Loss of CTLH component MAEA impairs DNA repair and replication and leads to developmental delay

Soren Hough, Satpal Jhujh, Samah Awwad, Oliver Lewis, Simon Lam, John Thomas, Thorsten Mosler, Aldo Bader, Lauren Bartik, Shane McKee, Shivarajan Amudhavalli, Estelle Colin, Nadirah Damseh, Emma Clement, Pilar Cacheiro, Anirban Majumdar, Damian Smedley, Joel Fluss, Rosalinda Gianinni, Isabelle Thiffault, Guido Zagnoli-Vieira, Rimma Belotserkovskaya, Stephen Smerdon, Petra Beli, Yaron Galanty, Christopher Carnie, Grant Stewart, and Stephen Jackson

Corresponding author(s): Christopher Carnie (ChristopherJames.Carnie@med.uni-heidelberg.de), Stephen Jackson (spj13@cam.ac.uk), Grant Stewart (g.s.stewart@bham.ac.uk), Yaron Galanty (yaron.galanty@cruk.cam.ac.uk)

Review Timeline:

Submission Date:	6th May 25
Editorial Decision:	20th Jun 25
Revision Received:	16th Oct 25
Editorial Decision:	11th Nov 25
Revision Received:	20th Nov 25
Accepted:	24th Nov 25

Editor: Zeljko Durdevic

Transaction Report:

20th Jun 2025

Dear Dr. Carnie,

Thank you for the submission of your manuscript to EMBO Molecular Medicine, and please accept my apologies for the unusual delay in getting back to you. We have now received feedback from two of the three reviewers who agreed to evaluate your manuscript. As the referee #2 will unfortunately not be able to return his/her report in a timely manner, we prefer to make a decision now in order to avoid further delay in the process. Should referee #2 provide a report, we will send it to you, with the understanding that we will not ask for an additional revision.

As you will see from their reports pasted below, both referees recognize potential interest of the manuscript but also raise important concerns that should be addressed in a major revision. If you would like to discuss further the points raised by the referees, I am available to do so via email or video. Let me know if you are interested in this option.

We would welcome the submission of a revised version within three months for further consideration. Please let us know if you require longer to complete the revision.

I look forward to receiving your revised manuscript.

Yours sincerely,

Zeljko Durdevic

Zeljko Durdevic
Senior Editor
EMBO Molecular Medicine

We require:

2) Individual production quality figure files as .eps, .tif, .jpg (one file per figure). For guidance, download the 'Figure Guide PDF': (<https://www.embopress.org/page/journal/17574684/authorguide#figureformat>).

3) A .docx formatted letter INCLUDING the reviewers' reports and your detailed point-by-point responses to their comments. As part of the EMBO Press transparent editorial process, the point-by-point response is part of the Review Process File (RPF), which will be published alongside your paper.

4) A complete author checklist, which you can download from our author guidelines (<https://www.embopress.org/page/journal/17574684/authorguide#submissionofrevisions>). Please insert information in the checklist that is also reflected in the manuscript. The completed author checklist will also be part of the RPF.

6) It is mandatory to include a 'Data Availability' section after the Materials and Methods. Before submitting your revision, primary datasets produced in this study need to be deposited in an appropriate public database, and the accession numbers and database listed under 'Data Availability'. Please remember to provide a reviewer password if the datasets are not yet public (see <https://www.embopress.org/page/journal/17574684/authorguide#dataavailability>).

12) Author contributions: You will be asked to provide CRediT (Contributor Role Taxonomy) terms in the submission system. These replace a narrative author contribution section in the manuscript.

13) A Conflict of Interest statement should be provided in the main text.

14) Every published paper now includes a 'Synopsis' to further enhance discoverability. Synopses are displayed on the journal

webpage and are freely accessible to all readers. They include a short stand first (maximum of 300 characters, including space) as well as 2-5 one-sentences bullet points that summarizes the paper. Please write the bullet points to summarize the key NEW findings. They should be designed to be complementary to the abstract - i.e. not repeat the same text. We encourage inclusion of key acronyms and quantitative information (maximum of 30 words / bullet point). Please use the passive voice. Please attach these in a separate file or send them by email, we will incorporate them accordingly.

15) Include a Reagents and Tools Table as part of the Methods section, which can be downloaded from our author guidelines (<https://www.embopress.org/page/journal/17574684/authorguide#structuredmethods>)

***** Reviewer's comments *****

Referee #1 (Comments on Novelty/Model System for Author):

Medical impact is medium, because the authors study a very rare disorder.

Referee #1 (Remarks for Author):

This manuscript describes patients with mutations in the MAEA gene. The features of these patients are somewhat consistent with DNA replication stress. In parallel experiments, the MAEA gene is shown to have a role in the cellular response to DNA replication stress. Overall, this is a solid study that merits publication without revisions, except for addressing points 1 and 6.

Specific Points:

1. Fig. 1 shows the results of a CRISPR-Cas9 screen that identified genes encoding subunits of the CTLH complex, as genes whose loss leads to sensitivity to camptothecin. The results are clear. My only point is that Supplemental Table 3 does not show the results of all the genes that were targeted by the CRISPR-Cas9 knockout library. Can the authors provide all the results?
2. Fig. 2 identifies mutations associated with neurodevelopmental defects in humans in MAEA, a subunit of the CTLH complex. The functional defect of these mutations is then confirmed by expressing the mutant proteins in cells with knockout of the endogenous gene and testing for sensitivity to camptothecin. These results are clear.
3. Fig. 3 shows that loss of MAEA impairs RAD51 loading, but not end-resection. These results are also clear.
4. Fig. 4 shows that MAEA knockout cells are sensitive to agents that cause DNA replication stress, specifically hydroxyurea (HU), aphidicolin and an ATR inhibitor (ATRi). These results are also clear.
5. Fig. 5 shows that untreated MAEA knockout cells have decreased fork speed and increased fork stalling. After treatment with HU they have increased fork degradation, as well as decreased fork restart after HU wash-out. Similar results were obtained using fibroblasts from patients with MAEA mutations.
6. Table 1 shows pictures of patients with mutations in the MAEA gene. I presume that the authors have permission to use photographs of their patients. Yet, very soon photographs will be sufficient to identify individuals by AI using data from social media platforms. Should the pictures be shown? What do they add? Maybe the facial features should be cited, instead?

Referee #3 (Remarks for Author):

This study investigates the role of ubiquitin E3 ligases in the DNA damage response (DDR). Using a CRISPR-Cas9 knockout screen targeting E3 ligases, the authors identify the CTLH E3 ligase complex-focusing particularly on its subunit MAEA-as essential for handling single-ended DNA double-strand breaks (seDSBs). Furthermore, patient data reveal that mutations in MAEA are associated with neurodevelopmental disorders. Functional analyses indicate that MAEA deficiency impairs

homologous recombination (HR) and replication fork restart, likely by disrupting RAD51 loading at DNA damage sites. The study establishes a link between this defect, genomic instability, and developmental abnormalities.

I commend the authors for their comprehensive screening approach and for successfully integrating both clinical and mechanistic insights. However, several aspects require revision or clarification:

Cell viability should be assessed in MAEA knockout and hypomorphic cell lines, as well as in other gene knockout models, without camptothecin treatment. This will clarify whether MAEA loss alone affects cell viability in Figures 1D, 1E, and 1F.

The authors should provide a more detailed rationale for highlighting MAEA specifically over other CTLH complex components, such as WDR26, RMND5A, GID8, and RANBP9, within the manuscript.

To further support the link between MAEA and neurodevelopmental defects, the authors should validate their findings in primary murine neurons or in SH-SY5Y cells as an in vitro model system for the results shown in Figures 1D-F, 2C-F, 3D and 3G, and 4A-C.

In Figure 3, it is important to clarify whether MAEA knockout affects the expression levels of DNA repair-associated proteins regulated by RAD51, such as DNA2.

In Supplementary Figure 2B, it would be helpful to extend the immunoblot detecting MAEA to include the molecular weight of eGFP-MAEA. Similarly, the blot detecting eGFP should be shown to the size of eGFP-MAEA to confirm expression.

Providing clear explanations and additional data addressing these concerns will significantly strengthen the manuscript and further highlight the impact of the study's findings.

******* Reviewer's comments *********Referee #1 (Comments on Novelty/Model System for Author):**

Medical impact is medium, because the authors study a very rare disorder.

Referee #1 (Remarks for Author):

This manuscript describes patients with mutations in the MAEA gene. The features of these patients are somewhat consistent with DNA replication stress. In parallel experiments, the MAEA gene is shown to have a role in the cellular response to DNA replication stress. Overall, this is a solid study that merits publication without revisions, except for addressing points 1 and 6.

We thank the Reviewer for their highly supportive feedback on our manuscript and are pleased to offer specific responses to their comments below.

Specific Points:

1. Fig. 1 shows the results of a CRISPR-Cas9 screen that identified genes encoding subunits of the CTLH complex, as genes whose loss leads to sensitivity to camptothecin. The results are clear. My only point is that Supplemental Table 3 does not show the results of all the genes that were targeted by the CRISPR-Cas9 knockout library. Can the authors provide all the results?

We thank the reviewer for highlighting this omission at our end; we have provided the full set of NormZ scores across the sgRNA library (**Dataset EV1**).

2. Fig. 2 identifies mutations associated with neurodevelopmental defects in humans in MAEA, a subunit of the CTLH complex. The functional defect of these mutations is then confirmed by expressing the mutant proteins in cells with knockout of the endogenous gene and testing for sensitivity to camptothecin. These results are clear.

3. Fig. 3 shows that loss of MAEA impairs RAD51 loading, but not end-resection. These results are also clear.

4. Fig. 4 shows that MAEA knockout cells are sensitive to agents that cause DNA replication stress, specifically hydroxyurea (HU), aphidicolin and an ATR inhibitor (ATRi). These results are also clear.

5. Fig. 5 shows that untreated MAEA knockout cells have decreased fork speed and increased fork stalling. After treatment with HU they have increased fork degradation, as well as decreased fork restart after HU wash-out. Similar results were obtained using fibroblasts from patients with MAEA mutations.

6. Table 1 shows pictures of patients with mutations in the MAEA gene. I presume that the authors have permission to use photographs of their patients. Yet, very soon photographs will be sufficient to identify individuals by AI using data from social media platforms. Should the pictures be shown? What do they add? Maybe the facial features should be cited, instead?

All such informed consent has already been obtained. However, the reviewer makes an important point about the future implications of AI. As such, we have removed the photographs from the clinical table and have added a comment to the Data Availability

statement (**lines 469-473 in the revised manuscript**) stating that these are available from the corresponding authors upon reasonable request.

Referee #3 (Remarks for Author):

This study investigates the role of ubiquitin E3 ligases in the DNA damage response (DDR). Using a CRISPR-Cas9 knockout screen targeting E3 ligases, the authors identify the CTLH E3 ligase complex-focusing particularly on its subunit MAEA-as essential for handling single-ended DNA double-strand breaks (seDSBs). Furthermore, patient data reveal that mutations in MAEA are associated with neurodevelopmental disorders. Functional analyses indicate that MAEA deficiency impairs homologous recombination (HR) and replication fork restart, likely by disrupting RAD51 loading at DNA damage sites. The study establishes a link between this defect, genomic instability, and developmental abnormalities.

I commend the authors for their comprehensive screening approach and for successfully integrating both clinical and mechanistic insights. However, several aspects require revision or clarification:

We thank the reviewer for their encouraging feedback on our manuscript, which has helped to enhance our study and expand the applicability of its findings. We offer our point-by-point responses to individual comments below:

Cell viability should be assessed in MAEA knockout and hypomorphic cell lines, as well as in other gene knockout models, without camptothecin treatment. This will clarify whether MAEA loss alone affects cell viability in Figures 1D, 1E, and 1F.

We have added representative images of untreated cells from clonogenic assays in MAEA-deficient U2OS and HAP1 cells, in which we observed that, in comparison to WT counterparts, MAEA knockout compromises colony formation and cell growth in U2OS cells but not in HAP1 cells (**new Fig. EV1E,F** and below). This could be consistent with the DNA replication defects we have observed in MAEA-deficient cells. Given that stable KO cell lines might have adapted somewhat to MAEA loss, we further tested clonogenicity of U2OS cells upon transient siRNA-mediated MAEA depletion and also observed reduced colony size and plating efficiency (**new Fig. EV1G-K** and below). These data are discussed in **lines 119-123** in the revised manuscript.

In summary, it seems that depending on the cell line, MAEA loss alone can compromise cell viability. This is consistent with our difficulties in conducting viability assays upon MAEA depletion in SH-SY5Y cells (expanded upon below), and data from Depmap showing that approximately 23% of cell lines are dependent on MAEA.

Fig. EV1

New Fig. EV1. (E-F) Representative images of untreated WT and MAEA-deficient U2OS (E) and HAP1 (F) cells. (G) Immunoblot for MAEA in U2OS following siRNA transfection. (H-I) Representative images (H) and colony counts (I) of U2OS cells with or without siRNA-mediated MAEA depletion. (J) Average colony size of siRNA transfected U2OS cells at 500 cells per well. (K) Total area coverage of U2OS colonies following siRNA transfection. (I-K) Bars represent the mean \pm SEM.

The authors should provide a more detailed rationale for highlighting MAEA specifically over other CTLH complex components, such as WDR26, RMND5A, GID8, and RANBP9, within the manuscript.

MAEA is the CTLH component known to possess a functional RING domain that is required for the CTLH to ubiquitylate its target proteins. Given that phenotypes arising from MAEA loss likely reflect dysregulation of ubiquitylation substrates of the CTLH complex, we focussed on MAEA as the functional 'core' of the complex. We have made this reasoning clearer in the text (see lines 102-103 in the revised manuscript). In addition, our identification of a new human disease associated with pathogenic variants in MAEA provided an ideal opportunity to study the clinical implications of the results of our CRISPR screen.

To further support the link between MAEA and neurodevelopmental defects, the authors should validate their findings in primary murine neurons or in SH-SY5Y cells as an in vitro model system for the results shown in Figures 1D-F, 2C-F, 3D and 3G, and 4A-C.

As suggested, we have expanded our study to examine the effects of MAEA loss on neuronal cells. The neurodevelopmental features of DIADEM might suggest an effect of MAEA loss on pre-neuronal cells, since fully differentiated neurons are post-mitotic and will not undergo DNA replication or DNA repair by RAD51-dependent homologous recombination. With that in mind, we have conducted these experiments in SH-SY5Y cells rather than primary mouse neurons.

To perform **cell viability assays** (corresponding to Figures 1D-F, 2C-F, and 4A-C), we first tried CRISPR-mediated depletion of MAEA from SH-SY5Y cells. Unfortunately, we were unable to generate monoclonal *MAEA*^{-/-} SH-SY5Y cells, perhaps suggesting that MAEA loss compromises SH-SY5Y cells' fitness, in line with our findings detailed in response to the Reviewer's above question. Therefore, we next used CRISPR-Cas9 to deplete MAEA transiently in a polyclonal population, seeding clonogenic survival assays at short timepoints after targeting and selection with puromycin. However, the plating efficiency of these targeted cells was extremely poor and made clonogenic survival assays impossible. Finally, we used siRNA to deplete in SH-SY5Y cells and subjected the cells to alamarBlue viability assays. Unfortunately, MAEA depletion appeared to drastically alter the metabolism of SH-SY5Y cells, resulting in accelerated metabolism of the alamarBlue reagent and preventing this assay from being used (**see below**). This surprising finding might reflect the role of MAEA in regulating cellular metabolism (Yi et al, *Mol Cell* 2024, Pham et al *Cell Rep* 2025). Alternatively, it may be the case that this cell type cannot tolerate compromised RAD51 loading even in unperturbed conditions.

Rev. Fig. 1

Rev. Fig. 1. AlamarBlue cell proliferation assay in SH-SY5Y cells with siLuc or siMAEA as indicated. MAEA-depleted cells metabolically reduced resazurin to fluorescent resorufin before control cells had performed the equivalent reaction to a sufficient extent to be distinguishable from the media only control.

However, using **immunofluorescence-based assays** (corresponding to Figures 3D,G), we were able to show that siRNA-mediated depletion of MAEA in SH-SY5Y cells recapitulated the cellular phenotypes previously observed in U2OS cells. Specifically, we observed that siMAEA in SH-SY5Y cells did not compromise DNA end resection following camptothecin treatment as measured using native BrdU staining for ssDNA, but caused a dramatic

reduction in RAD51 loading at seDSBs/damaged replication forks (**new Fig. EV4A-E** in the revised manuscript and below). These data are discussed in **lines 232-234** in the revised manuscript.

On balance, while we have not successfully recapitulated our cellular sensitivity phenotypes in SH-SY5Y cells due to technical or biological challenges, our finding that the key mechanistic basis of these phenotypes – impaired RAD51 loading – does recapitulate in these cells. This is strongly suggestive of a consistent source of genome instability across cell types. The observation that MAEA loss appears to have more severe consequences for cell fitness in SH-SY5Y cells could be consistent with the neurological basis of DIADEM. We will therefore pursue these ideas in follow-up studies and thank the reviewer for the suggestion that led us to these observations.

Fig. EV4

New Fig. EV4. (A) Immunoblot for MAEA depletion in polyclonal SH-SY5Y cells following CRISPR-Cas9 editing. (B) Quantification and (C) representative images of BrdU in SH-SY5Y cells treated with DMSO (1h) or camptothecin (1 μ M, 1 h). (D) Quantification and (E) representative images of RAD51 foci in S phase SH-SY5Y cells treated with DMSO or camptothecin. Bars in B and D represent median and interquartile range. P-values were generated by performing a two-tailed Kruskal-Wallis test. γ is a measure of effect size. Data

represents three independent experiments. * $P \leq 0.05$, ** $P \leq 0.01$, *** $P \leq 0.001$, **** $P \leq 0.0001$. Scale bars are 10 μm . For **B-C**, the top and bottom 1% of nuclei according to BrdU intensity were removed from the analysis to remove artifacts.

In Figure 3, it is important to clarify whether MAEA knockout affects the expression levels of DNA repair-associated proteins regulated by RAD51, such as DNA2.

To assess this, we have as suggested tested the effects of MAEA on expression levels of relevant homologous recombination-related proteins in MAEA-deficient cells in addition to RAD51, which we reported was unchanged in MAEA^{-/-} cells in the original manuscript. We have now tested protein levels of the pro-resection factors CtIP, EXO1 and DNA2, as well as the major proteins known to promote RAD51 loading (BRCA1, BRCA2, PALB2 and RAD51 itself), and have not found any alterations in the expression level of any of these proteins - see below and **new Fig. EV3F** in the revised manuscript. We have described these findings in **lines 230-232** in the revised manuscript.

Figure EV3

New Fig. EV3. (F) Immunoblot for indicated proteins in U2OS WT or MAEA KO cells.

In Supplementary Figure 2B, it would be helpful to extend the immunoblot detecting MAEA to include the molecular weight of eGFP-MAEA. Similarly, the blot detecting eGFP should be shown to the size of eGFP-MAEA to confirm expression.

As requested, we have provided an updated image in **new Fig. EV2** that now shows the sizes of both MAEA and eGFP-MAEA together

Providing clear explanations and additional data addressing these concerns will significantly strengthen the manuscript and further highlight the impact of the study's findings.

11th Nov 2025

Dear Dr. Carnie,

Thank you for the submission of your revised manuscript to EMBO Molecular Medicine. I am pleased to inform you that we will be able to accept your manuscript pending the following final amendments:

- 1) Authors: E-mail correspondence to John Thomas and Guido Zagnoli Vieira could not be delivered. Please update their e-mail addresses and make sure to enter correct e-mail addresses for all authors in our submission system.
- 2) In the main manuscript file, please do the following:
 - Please address all comments suggested by our data editors listed below:
 - o Figure legends:
 1. Please note that the exact p values are not provided in the legends of figures 3A, B, D, G; 4G, 5A, B, C, D, E, F; EV3 B, EV4 B, D.
 - There is a callout for Supplemental Figure 3C, please update.
 - Please remove Reagents and Tools Table from the manuscript file and uploaded it as a separate file. More information on how to adhere to this format as well as downloadable templates (.docx) for the Reagents and Tools Table can be found in our author guidelines: <https://www.embopress.org/page/journal/17574684/authorguide#structuredmethods>An example of a paper with Structured Methods can be found here:
<https://www.embopress.org/doi/full/10.1038/s44320-024-00037-6#sec-4>
 - Author contributions: Please remove it from the manuscript and specify author contributions in our submission system. CRediT has replaced the traditional author contributions section because it offers a systematic machine-readable author contributions format that allows for more effective research assessment. You are encouraged to use the free text boxes beneath each contributing author's name to add specific details on the author's contribution. More information is available in our guide to authors:
<https://www.embopress.org/page/journal/17574684/authorguide#authorshipguidelines>
 - In the Figure EV3 legend please correct the reference to reused images Fig. 2C, E to Fig. 3C, E.
 - Indicate in legends exact n and exact p values, not a range, along with the statistical test used. To keep the figures "clear" some authors found providing an Appendix table Sx with all exact p-values preferable. You are welcome to do this if you want to.
 - In data availability statement replace current text with the following sentence "This study includes no data deposited in external repositories."
- 3) Tables: Please upload EV Tables with their legends as separate excel files.
- 4) Source data: Please upload a completed source data checklist.
- 5) Funding: Please make sure that information about all sources of funding are complete in both our submission system and in the manuscript. Currently, RUK RadNET Cambridge C17918/A2887 is missing in our submission system. Please correct.
- 6) The Paper Explained. Please add it to the main manuscript file.
- 7) Synopsis:
 - Synopsis image: Please resize the image to 550 px-wide x 300-600 pixels high and upload it as a high-resolution jpeg file.
 - Please check your synopsis text and image before submission with your revised manuscript. Please be aware that in the proof stage minor corrections only are allowed (e.g., typos).
- 8) As part of the EMBO Publications transparent editorial process (see our Editorial at <http://embomolmed.embopress.org/content/2/9/329>), EMBO Molecular Medicine will publish online a Review Process File (RPF) to accompany accepted manuscripts. This file will be published in conjunction with your paper and will include the anonymous referee reports, your point-by-point response and all pertinent correspondence relating to the manuscript. Let us know whether you agree with the publication of the RPF and as here, if you want to remove or not any figures from it prior to publication. Please note that the Authors checklist will be published at the end of the RPF.
- 9) Please provide a point-by-point letter INCLUDING my comments as well as the reviewer's reports and your detailed responses (as Word file).

I look forward to reading a new revised version of your manuscript as soon as possible.

Yours sincerely,

Zeljko Durdevic

Zeljko Durdevic
Senior Editor
EMBO Molecular Medicine

*** Instructions to submit your revised manuscript ***

- 1) a .docx formatted version of the manuscript text (including Figure legends and tables)
 - 2) Separate figure files*
 - 3) supplemental information as Expanded View and/or Appendix. Please carefully check the authors guidelines for formatting Expanded view and Appendix figures and tables at <https://www.embopress.org/page/journal/17574684/authorguide#expandedview>
 - 4) a letter INCLUDING the reviewer's reports and your detailed responses to their comments (as Word file).
 - 5) The paper explained: EMBO Molecular Medicine articles are accompanied by a summary of the articles to emphasize the major findings in the paper and their medical implications for the non-specialist reader. Please provide a draft summary of your article highlighting
 - the medical issue you are addressing,
 - the results obtained and
 - their clinical impact.This may be edited to ensure that readers understand the significance and context of the research. Please refer to any of our published articles for an example.
 - 6) Author contributions: the contribution of every author must be detailed in our submission system.
 - 7) EMBO Molecular Medicine now requires a complete author checklist (<https://www.embopress.org/page/journal/17574684/authorguide>) to be submitted with all revised manuscripts. Please use the checklist as guideline for the sort of information we need WITHIN the manuscript. The checklist should only be filled with page numbers where the information can be found. This is particularly important for animal reporting, antibody dilutions (missing) and exact values and n that should be indicated instead of a range.
 - 8) Every published paper now includes a 'Synopsis' to further enhance discoverability. Synopses are displayed on the journal webpage and are freely accessible to all readers. They include a short stand first (maximum of 300 characters, including space) as well as 2-5 one sentence bullet points that summarise the paper. Please write the bullet points to summarise the key NEW findings. They should be designed to be complementary to the abstract - i.e. not repeat the same text. We encourage inclusion of key acronyms and quantitative information (maximum of 30 words / bullet point). Please use the passive voice. Please attach these in a separate file or send them by email, we will incorporate them accordingly.
- You are also welcome to suggest a striking image or visual abstract to illustrate your article. If you do please provide a jpeg file 550 px-wide x 300-600px high.
- 9) A Conflict of Interest statement should be provided in the main text
 - 10) Please note that we now mandate that all corresponding authors list an ORCID digital identifier. This takes <90 seconds to complete. We encourage all authors to supply an ORCID identifier, which will be linked to their name for unambiguous name identification.

Currently, our records indicate that the ORCID for your account is 0000-0001-7006-5596.

Please click the link below to modify this ORCID:
Link Not Available

11) Include a Reagents and Tools Table as part of the Methods section, which can be downloaded from our author guidelines (<https://www.embopress.org/page/journal/17574684/authorguide#structuredmethods>)

Photos 400-800 DPI

*Additional important information regarding figures and illustrations can be found at <https://bit.ly/EMBOPressFigurePreparationGuideline>. See also figure legend preparation guidelines: <https://www.embopress.org/page/journal/17574684/authorguide#figureformat>

***** Reviewer's comments *****

Referee #1 (Comments on Novelty/Model System for Author):

The disease being studied is very rare. This limits medical impact. However, the molecular mechanisms being studied are quite important.

Referee #1 (Remarks for Author):

The authors have addressed all my comments in this revised version of the manuscript.

Referee #3 (Remarks for Author):

Successfully revised the manuscript by substantiating our findings with multiple supporting experiments in response to reviewer feedback.

Point-by-point reply and summary of changes

1) Authors: E-mail correspondence to John Thomas and Guido Zagnoli Vieira could not be delivered. Please update their e-mail addresses and make sure to enter correct e-mail addresses for all authors in our submission system. Done

2) In the main manuscript file, please do the following:

- Please address all comments suggested by our data editors listed below:

o Figure legends:

1. Please note that the exact p values are not provided in the legends of figures 3A, B, D, G; 4G, 5A, B, C, D, E, F; EV3 B, EV4 B, D. As suggested below, we have included exact values in Appendix Table S1 and added callouts to this table in the relevant legends. Note that during this annotation, we noted a minor display error in Fig. 5D, in which two asterisks were missing from the WT vs E349K comparison. We have rectified this and uploaded a new PDF of this figure.

- There is a callout for Supplemental Figure 3C, please update. Updated to Fig. EV3C-D

- Please remove Reagents and Tools Table from the manuscript file and uploaded it as a separate file. More information on how to adhere to this format as well as downloadable templates (.docx) for the Reagents and Tools Table can be found in our author guidelines: https://www.embopress.org/page/journal/17574684/authorguide#structured_methods

An example of a paper with Structured Methods can be found here:

<https://www.embopress.org/doi/full/10.1038/s44320-024-00037-6#sec-4>

Done

- Author contributions: Please remove it from the manuscript and specify author contributions in our submission system. CRediT has replaced the traditional author contributions section because it offers a systematic machine-readable author contributions format that allows for more effective research assessment. You are encouraged to use the free text boxes beneath each contributing author's name to add specific details on the author's contribution. More information is available in our guide to authors:

<https://www.embopress.org/page/journal/17574684/authorguide#authorshipguidelines>

Done

- In the Figure EV3 legend please correct the reference to reused images Fig. 2C, E to Fig. 3C, E.

Done

- Indicate in legends exact n and exact p values, not a range, along with the statistical test used. To keep the figures "clear" some authors found providing an Appendix table Sx with all exact p-values preferable. You are welcome to do this if you want to.

Done – as suggested, we have included an Appendix Table S1 for conveying exact P-values and added callouts in the relevant legends.

- In data availability statement replace current text with the following sentence "This study includes no data deposited in external repositories." Done

3) Tables: Please upload EV Tables with their legends as separate excel files. Done

4) Source data: Please upload a completed source data checklist. Done

5) Funding: Please make sure that information about all sources of funding are complete in both our submission system and in the manuscript. Currently, RUK RadNET Cambridge C17918/A2887 is missing in our submission system. Please correct. Done

6) The Paper Explained. Please add it to the main manuscript file. Done

7) Synopsis:

- Synopsis image: Please resize the image to 550 px-wide x 300-600 pixels high and upload it as a high-resolution jpeg file. Done

- Please check your synopsis text and image before submission with your revised manuscript. Please be aware that in the proof stage minor corrections only are allowed (e.g., typos). Done

8) As part of the EMBO Publications transparent editorial process (see our Editorial at <http://embomolmed.embopress.org/content/2/9/329>), EMBO Molecular Medicine will publish online a Review Process File (RPF) to accompany accepted manuscripts. This file will be published in conjunction with your paper and will include the anonymous referee reports, your point-by-point response and all pertinent correspondence relating to the manuscript. Let us know whether you agree with the publication of the RPF and as here, if you want to remove or not any figures from it prior to publication. Please note that the Authors checklist will be published at the end of the RPF. We agree with this publication including the additional figures in our response to reviewers.

9) Please provide a point-by-point letter INCLUDING my comments as well as the reviewer's reports and your detailed responses (as Word file). Done

***** Reviewer's comments *****

Referee #1 (Comments on Novelty/Model System for Author):

The disease being studied is very rare. This limits medical impact. However, the molecular mechanisms being studied are quite important.

Referee #1 (Remarks for Author):

The authors have addressed all my comments in this revised version of the manuscript.

We thank the reviewer for taking the time to review our manuscript and for their constructive feedback.

Referee #3 (Remarks for Author):

Successfully revised the manuscript by substantiating our findings with multiple supporting experiments in response to reviewer feedback.

We thank the reviewer for taking the time to review our manuscript and for their constructive feedback.

24th Nov 2025

Dear Dr. Carnie,

We are pleased to inform you that your manuscript is accepted for publication and is now being sent to our publisher to be included in the next available issue of EMBO Molecular Medicine.

Zeljko Durdevic
Senior Editor
EMBO Molecular Medicine
